# Provably Efficient Off-Policy Adversarial Imitation Learning with Convergence Guarantees

**Yilei Chen**                                                                    *ylchen9@bu.edu*
*Division of Systems Engineering*
*Boston University*
*Boston, MA 02215, USA*

**Vittorio Giammarino**                                                           *vgiammar@purdue.edu*
*Department of Computer Science*
*Purdue University*
*West Lafayette, IN 47907, USA*

**James Queeney**                                                                 *jqueeney@amazon.com*
*Amazon Robotics*
*North Reading, MA 01864, USA*

**Ioannis Ch. Paschalidis**                                                       *yannisp@bu.edu*
*Department of Electrical and Computer Engineering*
*Boston University*
*Boston, MA 02215, USA*

**Reviewed on OpenReview:** *https://openreview.net/forum?id=OahvMeRgKP*

## Abstract

*Adversarial Imitation Learning (AIL)* faces challenges with sample inefficiency because of its reliance on sufficient on-policy data to evaluate the performance of the current policy during reward function updates. In this work, we study the convergence properties and sample complexity of off-policy AIL algorithms. We show that, even in the absence of importance sampling correction, reusing samples generated by the $o(\sqrt{K})$ most recent policies, where $K$ is the number of iterations of policy updates and reward updates, does not undermine the convergence guarantees of this class of algorithms. Furthermore, our results indicate that the distribution shift error induced by off-policy updates is dominated by the benefits of having more data available. This result provides theoretical support for the sample efficiency of off-policy AIL algorithms that has been observed in practice.

## 1 Introduction

In *Imitation Learning (IL)* (Osa et al., 2018), agents do not have access to reward feedback. Instead, they rely on a set of trajectories generated by an expert's policy and the primary objective is to train a policy that achieves performance comparable to that policy. *Adversarial Imitation Learning (AIL)* (Ho & Ermon, 2016; Fu et al., 2018) has emerged as a popular approach for IL. AIL frames the IL problem as a repeated two-player game. In each iteration, an adversary updates the reward function to widen the gap between expert and agent performance, while the agent updates its policy to narrow this gap.

In order to perform reward updates at every iteration, the standard AIL objective requires samples generated from the agent's current policy (i.e., on-policy data) in order to evaluate the policy's expected cumulative rewards. The use of on-policy data for reward updates has been critical for establishing convergence guarantees of AIL algorithms, but represents a main limitation of these approaches. Specifically, on-policy AIL algorithms require a significant number of new interactions between the agent and the environment at ev-

ery update, precluding the use of these algorithms in settings where interactions with the environment are expensive or limited.

To relax this on-policy requirement, off-policy methods have been proposed to enhance the sample efficiency of AIL algorithms (Kostrikov et al., 2018; Sasaki et al., 2018). These methods reuse samples from previous policies (i.e., off-policy data) during reward updates, which improves efficiency but also introduces a distribution shift error. As a result, the use of off-policy data alters the standard objective used for reward updates in on-policy AIL. Due to this change in the objective for reward updates, the theoretical properties of off-policy AIL algorithms are not well understood.

In the off-policy AIL setting, the main challenge is the data distribution shift induced by off-policy data. Instead of applying off-policy correction techniques such as importance sampling or variable transformations (Kostrikov et al., 2019), we are interested in answering the following question: *can we guarantee the convergence of off-policy AIL by controlling the distribution shift error introduced by off-policy data?* In this work, we show this is possible by reusing samples generated by the $o(\sqrt{K})$ most recent policies when computing reward updates, where $K$ represents the total number of updates. Using off-policy data in this way, we combine off-policy projected gradient ascent reward updates with model-based mirror descent policy updates. By doing so, we make the following main contributions:

1. We provide convergence guarantees for an off-policy AIL algorithm that does not require off-policy correction techniques.

2. We provide theoretical support for the sample efficiency of our off-policy AIL algorithm. Our results shed light on how the size of the state space, action space, and horizon length can affect the choice of data during reward updates.

3. In addition to our theoretical analysis, we demonstrate the practical performance of our algorithm through experiments on both discrete space MiniGrid tasks (Chevalier-Boisvert et al., 2023) and continuous space OpenAI Gym tasks (Brockman et al., 2016).

## 2 Related Work

Numerous studies have considered the theoretical properties of AIL, focusing mainly on the on-policy setting. Specifically, Cai et al. (2019) first showed the global convergence of AIL in the special linear quadratic regulator setting. Chen et al. (2019) studied the convergence of AIL to a stationary point, as opposed to global convergence, within a general Markov Decision Process framework. Guan et al. (2021) analyzed the global convergence of AIL when paired with different policy update algorithms. Xu et al. (2020) and Zhang et al. (2020) studied the global convergence of AIL with neural networks in the tabular and continuous case, respectively. Liu et al. (2022b) proposed AIL algorithms with global convergence guarantees with linear function approximations. Shani et al. (2022) proposed a solution via two mirror descent-based no-regret algorithms. Xu et al. (2023) connected reward-free exploration and AIL and achieved the minimax optimal expert sample complexity and interaction complexity. Note that all of these works consider on-policy AIL methods. Compared to these works, our research focuses on providing convergence guarantees for off-policy AIL, where off-policy data are used for reward updates.

Another line of research has focused on designing off-policy AIL algorithms, and existing works have demonstrated the strong practical performance of off-policy AIL. Kostrikov et al. (2018) suggested an algorithm that enhances sample efficiency by directly using off-policy data. Giammarino et al. (2023) and Liu et al. (2022a) leveraged off-policy AIL as in Kostrikov et al. (2018) for the problem of imitation from expert videos. Sasaki et al. (2018) developed a method based on supervised classification of expert and agent transitions. Despite the practical performance of these algorithms, none of them provided theoretical convergence guarantees. The use of off-policy data modifies the on-policy AIL objective for reward updates, thereby forfeiting the strong theoretical guarantees of on-policy AIL methods.

The alteration of the AIL objective in the off-policy setting can be addressed by introducing *Importance Sampling (IS)* corrections. However, note that IS induces high variance (Liu et al., 2018) during the policy

evaluation step, which results in the need for more interactions with the environment in order to accurately evaluate the policy. Beyond importance sampling, Kostrikov et al. (2019) introduced an off-policy algorithm for AIL by providing a new representation of the divergence objective that avoids the use of any explicit on-policy expectations. It yields a completely off-policy objective which is the same as the original AIL objective, but this new objective is obtained by means of variable transformations (Nachum et al., 2019) and becomes more difficult to optimize in practice. Similar techniques were used by Hoshino et al. (2022) and Zhu et al. (2020) to formalize off-policy algorithms in learning from demonstrations and learning from observations settings, respectively. Different from these methods, our work focuses on establishing convergence guarantees for off-policy AIL without requiring off-policy correction techniques.

Beyond the scope of AIL, our work lies in the broader domain of imitation learning, which includes a wide range of other learning paradigms including behavioral cloning (BC) (Bain & Sammut, 1995; Ross et al., 2011; Daftry et al., 2016) and its variants with different forms of regularization to mitigate compounding errors (Seo et al., 2023); offline RL-based methods (Rashidinejad et al., 2021); preference-based imitation learning and direct occupancy-matching methods (Ma et al., 2022). Importantly, our contribution is orthogonal to these paradigms: rather than proposing a new imitation learning algorithm to compete with existing frameworks, our work aims to bridge the gap between theory and practice in AIL by establishing the convergence guarantees and sample complexity advantages of off-policy AIL algorithms.

## 3 Preliminaries

Unless indicated otherwise, we denote the set $\{a_1, a_2, \ldots, a_N\}$ by $\{a_n\}_{n=1}^N$ and $\{1, 2, \ldots, N\}$ by $[N]$. $\Delta(\mathcal{X})$ denotes the space of probability distributions over a set $\mathcal{X}$. $O(\cdot)$ hides logarithmic terms and constants. We write $\mathbb{P}(\cdot)$ for probability, $\mathbb{E}[\cdot]$ for expectation, and $\mathbb{E}_\pi[\cdot]$ for the expectation over the trajectories induced by $\pi$. Finally, we write the Total Variation (TV) distance between two distributions as $\mathbb{D}_{\text{TV}}(\cdot||\cdot)$ and the Kullback-Leibler (KL) divergence between two distributions as $\mathbb{D}_{\text{KL}}(\cdot||\cdot)$.

**Reinforcement learning.** In this work, we consider undiscounted finite-horizon Markov Decision Processes (MDPs) defined as the tuple $\{\mathcal{S}, \mathcal{A}, H, \{P_h\}_{h=1}^H, \nu_1, \{r_h\}_{h=1}^H\}$, where $\mathcal{S}$ is the state space, $\mathcal{A}$ the action space, $H$ is the episode horizon, $P_h(s_{h+1} \mid s_h, a_h) \in \Delta(\mathcal{S})$ is the state-transition distribution at step $h$, $\nu_1$ the initial state distribution, and $r_h(s_h, a_h) \in [0, 1]$ is the bounded reward function at step $h$. To simplify exposition and avoid unnecessary technicalities, we assume $\mathcal{S}$ and $\mathcal{A}$ are finite with respective cardinalities $S$ and $A$, although our results can be easily extended to linear MDPs (cf. Appendix). A policy $\pi = \{\pi_h\}_{h=1}^H$ is a mapping from states to a probability distribution over actions $\pi : \mathcal{S} \times [H] \to \Delta(\mathcal{A})$, with $\pi_h(a_h \mid s_h)$ the probability of choosing action $a_h$ in state $s_h$ at step $h$. We consider an episode as the trajectory $\{(s_h, a_h, r_h)\}_{h=1}^H$, where $s_1 \sim \nu_1$, $a_h \sim \pi_h(\cdot \mid s_h)$, and $s_{h+1} \sim P_h(\cdot \mid s_h, a_h)$.

Given a policy $\pi$, the state value function $V^\pi : \mathcal{S} \times [H] \to [0, H]$ and the state-action value function (i.e., Q function) $Q^\pi : \mathcal{S} \times \mathcal{A} \times [H] \to [0, H]$ represent the expected cumulative rewards obtained by following the policy $\pi$ from a given state and state-action pair, respectively, where we write $V_h^\pi(s) = \mathbb{E}_\pi \left[ \sum_{t=h}^H r_h(s_t, a_t) | s_h = s \right]$, $Q_h^\pi(s, a) = \mathbb{E}_\pi \left[ \sum_{t=h}^H r_h(s_t, a_t) | s_h = s, a_h = a \right]$.

We define the performance of a policy $\pi$ under the reward function $r$ as $J(\pi, r) = \mathbb{E}_{s_1 \sim \nu_1} [V_1^\pi(s_1)]$. Moreover, note that $Q_h^\pi(s, a) = r_h(s, a) + \sum_{s' \in \mathcal{S}} P_h(s' \mid s, a) V_{h+1}^\pi(s')$ and $V_h^\pi(s) = \sum_{a \in \mathcal{A}} \pi(a \mid s) Q_h^\pi(s, a)$, where $V_{H+1}^\pi(s) = 0$. Finally, we denote with $\nu_h^\pi(s) = \mathbb{P}(s_h = s | \nu_1, \pi, P)$ the state visitation distribution induced by the policy $\pi$ at step $h$ and with $d_h^\pi(s, a) = \nu_h^\pi(s) \pi_h(a|s)$ the state-action visitation distribution at step $h$.

**Adversarial imitation learning.** In AIL, we assume the agent has access to a set of $N_E$ trajectories generated by the expert's policy. The objective of AIL is to learn a policy with performance comparable to the expert within a reward function space. Formally, AIL considers the minimax problem

$$\min_{\pi \in \Pi} \max_{\mu \in \mathcal{R}} L(\pi, \mu) = J(\pi^E, r^\mu) - J(\pi, r^\mu), \tag{1}$$

where $\pi^E$ represents the expert's policy, $\Pi$ is a space of feasible policies for the agent, and $\mu = \{\mu_h\}_{h=1}^H$ is a parameterization of the reward function $r^\mu$ in a reward function space $\mathcal{R}$. Note that we consider a tabular

---

**Algorithm 1** Standard AIL Scheme

---

**Input:** $N_E$ expert trajectories
Initialize policy $\pi^0$ and reward $\mu^0$.
**for** $k = 1$ **to** $K$ **do**
    Collect $B$ trajectories by following $\pi^{k-1}$.
    # Reward Update
    Update $\mu^k$ by maximizing $L(\pi^{k-1}, \mu)$ over $\mu$.
    # Policy Update
    Update $\pi^k$ by maximizing $J(\pi, \mu^k)$ over $\pi$.
**end for**

---

parameterization of the reward function, i.e., $r^\mu = \mu \in \mathbb{R}^{S \times A}$. See the Appendix for a generalization of our results to a linear reward parameterization.

In this work, we consider the regret of an AIL algorithm as defined in Shani et al. (2022), which measures the difference in performance between the expert and agent throughout the learning process for the worst-case choice of the reward function in $\mathcal{R}$.

**Definition 3.1** (AIL Regret)**.** The regret of an AIL algorithm over $K$ updates is given by

$$\text{Regret}_{\text{AIL}} = \sup_{\mu \in \mathcal{R}} \sum_{k=1}^{K} L(\pi^k, \mu),$$

where $\pi^k$ is the policy of the agent at update $k \in [K]$.

Shani et al. (2022) showed that the regret of an AIL algorithm can be partitioned into two independent subproblems: policy updates and reward updates. This is formally stated in the following result.

**Lemma 3.2** (Lemma 2 in Shani et al. 2022)**.** *The regret of an AIL algorithm over $K$ updates can be bounded by*

$$\text{Regret}_{\text{AIL}} \leq \underbrace{\sup_{\pi \in \Pi} \sum_{k=1}^{K} J(\pi, \mu^k) - J(\pi^k, \mu^k)}_{\textit{Regret of Policy Updates}} + \underbrace{\sup_{\mu \in \mathcal{R}} \sum_{k=1}^{K} L(\pi^k, \mu) - L(\pi^k, \mu^k)}_{\textit{Regret of Reward Updates}}. \tag{2}$$

Lemma 3.2 motivates the iterative nature of the AIL framework. At iteration $k$, we first update the reward function by maximizing $L(\pi^{k-1}, \mu)$ over $\mu$. To that end, notice that we need to estimate $J(\pi^E, \mu)$ and $J(\pi^{k-1}, \mu)$ (cf. equation 1). $J(\pi^E, \mu)$ can be approximated based on the set of expert trajectories, while evaluating $J(\pi^{k-1}, \mu)$ requires new interactions with the environment by following $\pi^{k-1}$. After updating the reward function to $\mu^k$, we update the policy by maximizing $J(\pi, \mu^k)$ over $\pi$ (cf. equation 1). This is equivalent to solving a standard RL problem. We summarize this standard scheme for AIL algorithms in Algorithm 1.

**Remark.** To simplify the exposition, we assume the number of expert trajectories is infinite. Access to a finite number of expert trajectories would induce an additional statistical error, but has no impact on the focus of this work.

## 4  Off-Policy Adversarial Imitation Learning

Based on Lemma 3.2, in order to show the convergence of AIL algorithms, we need to design algorithms that achieve sublinear regret in both subproblems: reward updates and policy updates. The policy update problem is a standard RL problem where the reward function is available. In order to ensure sample efficiency for the RL problem, we will apply a model-based method used in previous works (Shani et al., 2022; Liu

et al., 2022b). Unlike policy updates, it is not clear how to reuse past data when performing reward updates in a way that still guarantees global convergence of the AIL algorithm. The main goal of reward updates in AIL is to distinguish between expert and agent policies by maximizing the loss

$$L(\pi^{k-1}, \mu) = J(\pi^E, \mu) - J(\pi^{k-1}, \mu),$$

where $\pi^{k-1}$ is the current policy. However, when we leverage off-policy data from the last $N$ policies in the reward update, we end up maximizing a different objective given by

$$J(\pi^E, \mu) - \frac{1}{N} \sum_{n=1}^{N} J(\pi^{k-n}, \mu).$$

Therefore, we have introduced distribution shift error into the objective of our reward updates, which we must control in order to provide convergence guarantees.

The off-policy setting requires a careful balance between policy and reward updates. Small policy updates are needed to successfully control the distribution shift error in the off-policy reward update subproblem, but these updates must be large enough to guarantee sublinear regret in the policy update subproblem as well. By carefully selecting the size of policy updates and the amount of past data to use, we can enable sample reuse during reward updates while retaining global convergence guarantees. Based on this observation, we propose an off-policy AIL algorithm with convergence guarantees by combining (i) a KL divergence regularized model-based policy update and (ii) an off-policy projected gradient ascent reward update.

For the succinct presentation of the main idea, we leave the proofs of the main theorems and lemmas in this section to the Appendix.

### 4.1 Convergent Off-Policy AIL

**Policy updates.** Our policy updates leverage a model-based mirror descent algorithm that has been adopted in previous works (Shani et al., 2022; Liu et al., 2022b). The algorithm consists of two steps. We first estimate the state-transition function $\{P_h\}_{h=1}^H$ by empirical state-transition encounters $\hat{P}_h(s' \mid s, a) = \frac{n_h(s,a,s')}{n_h(s,a)}$, where $n_h(s,a)$ and $n_h(s,a,s')$ count the number of visitations to state-action pair $(s,a)$ and state-action-state pair $(s,a,s')$, respectively, at step $h$. If $n_h(s,a)$ is zero, we assume that the transition function is uniform at $(s,a)$. With the estimated state-transition function, we can evaluate the Q functions by recursion from $h = H$ to $h = 1$ as

$$\hat{Q}_h^\pi(s,a) = \min\{\mu_h(s,a) + b_h(s,a) + \sum_{s' \in \mathcal{S}} \hat{P}_h(s' \mid s, a)\hat{V}_{h+1}^\pi(s'), H - h + 1\},$$

$$\hat{V}_h^\pi(s) = \sum_{a \in \mathcal{A}} \hat{Q}_h^\pi(s,a)\pi_h(a|s),$$

where $\hat{V}_{H+1}(s) = 0$ and $b_h$ is an optimistic UCB-bonus to encourage exploration (refer to Cai et al. 2020 for more details). Second, we update the policy by maximizing $J(\pi, \mu)$ using the KL divergence regularized mirror descent algorithm

$$\pi^k = \arg\max_{\pi \in \Pi}\{\langle \nabla_\pi J(\pi^{k-1}, \mu^k), \pi - \pi^{k-1}\rangle + J(\pi^{k-1}, \mu^k) - \sigma^{-1}D(\pi, \pi^{k-1})\}, \tag{3}$$

where $\sigma$ is the step size and $D(\pi, \pi^{k-1}) := \mathbb{E}_{s \sim d^{\pi^{k-1}}}[\mathbb{D}_{\mathrm{KL}}(\pi \| \pi^{k-1})[s]]$ is the expected KL divergence between $\pi$ and $\pi^{k-1}$. It can be shown that equation 3 has the closed-form solution

$$\pi_h^k(\cdot|s) \propto \pi_h^{k-1}(\cdot|s) \cdot \exp\{\sigma \cdot \hat{Q}_h^{\pi^{k-1}}(s, \cdot)\}. \tag{4}$$

Shani et al. (2022) showed that this algorithm achieves sublinear regret for the policy update subproblem, as described in the following result.

**Lemma 4.1** (Lemma 4 in Shani et al. 2022)**.** *Let $\sigma = \sqrt{2\log A/(H^2 K)}$. With probability at least $1 - \delta$, the regret of the policy update problem is bounded by*

$$\text{Regret}_\pi = \sup_{\pi \in \Pi} \sum_{k=1}^K J(\pi, \mu^k) - J(\pi^k, \mu^k) \leq \tilde{O}(\sqrt{H^4 S^2 A K}).$$

Note that the high-probability logarithm term in the above lemma is hidden in the notation $\tilde{O}$ and similarly for the subsequent lemmas. Lemma 4.1 shows that the policy update in equation 3 is aggressive enough to achieve sublinear policy regret. In the next part, we show how to achieve sublinear reward regret while using off-policy data by controlling the distribution shift error introduced by the policy update in equation 3.

If we step outside the AIL framework, the policy updates is simply an adversarial MDP problem in the episodic setting with unknown transitions and full information (i.e., the rewards of all state-action pairs are known at each episode). There are a bunch of other algorithms can achieve sub-linear regret in this setting, including Follow-the-Regularized-Leader (FTRL) (Jin et al., 2021), Follow-the-Perturbed-Leader (FPL) (Wang & Dong, 2020), and policy gradient methods (He et al., 2022). However, in the off-policy AIL setting where we will utilize past trajectories in each reward update, policy updates must not only achieve sub-linear regret but also satisfy a stability requirement—namely, that the updated policy remains close to the one in the last step, which we will elaborate on in the following part. This condition is important to avoid excessive distribution shift. Among the available algorithms, the one we adopt—based on optimism and mirror descent—achieves the current best-known upper bound on policy update regret while also guaranteeing a bound on the deviation between consecutive policies, as established in Theorem 4.3.

**Reward updates.** We update the reward parameters $\mu$ by maximizing $L(\pi, \mu)$ using projected gradient ascent. The on-policy version takes the form

$$\mu_h^k = \text{Proj}_{\mu \in \mathcal{R}} \left\{ \mu_h^{k-1} + \eta \nabla_{\mu_h} L(\pi^{k-1}, \mu^{k-1}) \right\},$$

where $\eta$ is the step size. By definition, the objective function $L(\pi, \mu)$ and the state-action visitation distribution have the relationship: $L(\pi, \mu) = \sum_{h=1}^H \langle d_h^{\pi^E} - d_h^\pi, \mu_h \rangle$, $\nabla_{\mu_h} L(\pi, \mu) = d_h^{\pi^E} - d_h^\pi$.

As previously mentioned, the reward update process in the on-policy setting is not sample efficient, as it requires a significant number of new interactions between the agent and the environment to evaluate $L(\pi^{k-1}, \mu^{k-1})$ at every iteration. Leveraging off-policy data is a practical way to make the reward update process more efficient.

In order to achieve efficiency while guaranteeing convergence at the same time, we propose an off-policy algorithm that considers data from only the $N(k)$ most recent policies at round $k$. Compared to on-policy AIL which considers data sampled from $d_h^{\pi^{k-1}}$, our off-policy approach samples data from the distribution $d_h^{k-1,mix} = \sum_{n=1}^N \beta_n d_h^{\pi^{k-n}}$, where $\{\beta_n\}_{n=1}^N$ is a distribution which parameterizes the sample weights of the $N$ policies. In principle, $\{\beta_n\}_{n=1}^N$ can be any distribution and it recovers the on-policy setting when $\{\beta_n\}_{n=1}^N = \{1, 0, \ldots, 0\}$. Throughout this work, we assume $\{\beta_n\}_{n=1}^N$ is a uniform distribution and we leave the problem of finding the optimal distribution to future works. The off-policy reward update algorithm takes the form

$$\mu_h^k = \text{Proj}_{\mu \in \mathcal{R}} \left\{ \mu_h^{k-1} + \eta \nabla_{\mu_h} L(\pi^{k-1,mix}, \mu^{k-1}) \right\} \tag{5}$$

where $L(\pi^{k-1,mix}, \mu) = J(\pi^E, \mu) - \frac{\sum_{n=1}^N J(\pi^{k-n}, \mu)}{N}$.

Notice that the off-policy reward update performs a gradient step on the objective $L(\pi^{k-1,mix}, \mu)$, compared to the on-policy reward update which considers the objective $L(\pi^{k-1}, \mu)$. Therefore, we must consider the distribution shift error introduced by altering the reward update objective. Achiam et al. (2017) proved that in infinite-horizon discounted MDPs, the difference between two state visitation distributions can be bounded by the total variation distance between the corresponding policies: $||\nu^\pi - \nu^{\pi'}||_1 \leq \frac{2\gamma}{1-\gamma} \mathbb{E}_{s \sim d^\pi} [\mathbb{D}_{\text{TV}}(\pi||\pi')[s]]$, where $\gamma$ is the discount factor. In our work, we extend this result to finite-horizon undiscounted MDPs.

**Lemma 4.2.** *In finite-horizon undiscounted MDPs, the divergence between state-action visitation distributions is bounded by the total variation distance of the corresponding policies according to*

$$||d_h^\pi - d_h^{\pi'}||_1 \leq 2\sum_{i=1}^h \mathbb{E}_{s\sim\nu_i^\pi}[\mathbb{D}_{\text{TV}}(\pi_i||\pi_i')[s]], \ \forall h \in [H].$$

By Lemma 4.2, in order to bound the off-policy data distribution shift during reward updates, the policy update algorithm should limit the total variation distance between consecutive policies. In the following result, we show that the policy update in equation 3 is conservative enough to accomplish this goal.

**Theorem 4.3.** *The total variation distance between two consecutive policies induced by implementing the policy update algorithm in equation 3 is bounded by*

$$\mathbb{D}_{\text{TV}}(\pi_h^k||\pi_h^{k-1})[s] \leq O(AH\sigma), \ \forall s \in \mathcal{S}, k \in [K], h \in [H].$$

*When $\sigma = \sqrt{2\log A/(H^2 K)}$, the bound becomes $O(\sqrt{2A^2\log A/K})$.*

Based on Lemma 4.2 and Theorem 4.3, we can show that the reward update in equation 5 achieves sublinear regret for the reward update problem, despite altering the objective function.

**Theorem 4.4.** *Let $\eta = \sqrt{SA/K}$ and $\sigma = \sqrt{2\log A/(H^2 K)}$. Then, with probability at least $1 - \delta$, the regret of the reward update problem is bounded by*

$$Regret_\mu = \sup_{\mu\in\mathcal{R}} \sum_{k=1}^K L(\pi^k, \mu) - L(\pi^k, \mu^k)$$

$$\leq \tilde{O}\Big(\underbrace{\sqrt{H^2 SAK}}_{\text{regret of reward updates}} + \underbrace{\sqrt{\frac{H^4 A^2(\sum_k(N(k)-1))^2}{K}}}_{\text{error induced by off-policy updates}} + \underbrace{\sqrt{\frac{H^3 SAK^2}{B\sum_k N(k)}}}_{\text{estimation error}}\Big),$$

*where $B$ is the number of new trajectories collected with the current policy between updates. Specifically, if $N$ is fixed, we have*

$$Regret_\mu \leq \tilde{O}\Big(\underbrace{\sqrt{H^2 SAK}}_{\text{regret of reward updates}} + \underbrace{\sqrt{H^4 A^2(N-1)^2 K}}_{\text{error induced by off-policy updates}} + \underbrace{\sqrt{H^3 SAK/(BN)}}_{\text{estimation error}}\Big).$$

**Main result.** Based on the regret bound for the policy update problem in Lemma 4.1 and the regret bound for the reward update problem in Theorem 4.4, we are now ready to state our main theoretical result for the off-policy AIL algorithm.

**Theorem 4.5.** *In the off-policy AIL algorithm based on the policy update in equation 3 and the reward update in equation 5, let $\sigma = \sqrt{2\log A/(H^2 K)}$ and $\eta = \sqrt{SA/K}$. Then, with probability at least $1 - \delta$, it holds that*

$$Regret_{AIL} \leq \tilde{O}\Big(\underbrace{\sqrt{H^4 S^2 AK}}_{\text{regret of policy updates}} + \underbrace{\sqrt{H^2 SAK}}_{\text{regret of reward updates}}$$

$$+ \underbrace{\sqrt{H^4 A^2(N-1)^2 K}}_{\text{error induced by off-policy updates}} + \underbrace{\sqrt{H^3 SAK/(BN)}}_{\text{estimation error}}\Big).$$

*Proof.* It can be derived by plugging Lemma 4.1 and Theorem 4.4 into the regret of policy updates and reward updates, respectively, in Lemma 3.2. $\qquad\square$

Based on Theorem 4.5, we can directly derive the following two propositions.

**Proposition 4.6.** *In the case where $N$ is fixed, the regret bound in Theorem 4.4 is optimized when we update reward parameters using data from the $N^* = \tilde{O}(\frac{S}{HAB})^{1/3}$ most recent policies.*

**Proposition 4.7.** *The regret bound in Theorem 4.4 is sublinear with respect to $K$ when we update reward parameters using data from the $N(k) = o(\sqrt{K}), \forall k \in [K]$ most recent policies, including the case where $N$ is a fixed number.*

On-policy AIL (i.e., AIL using on-policy reward updates) is a special case in our analysis where $N = 1$. Proposition 4.6 shows that there is an optimal scaling for the number of recent policies to consider during the reward updates based on the worst-case regret. In the cases where $S$ is dominant, the off-policy AIL algorithm is expected to have better performance by reusing data from prior policies. In the cases where $H$ and $A$ are dominant, it is better to only consider data from the current policy during reward updates.

### 4.2 Sample Efficient Off-Policy AIL

In this section, we illustrate why our off-policy AIL algorithm can be more sample efficient than the on-policy AIL algorithm.

The set of feasible state-action visitation distributions is defined as $\mathcal{D} = \{d = \{d_h\}_{h=1}^H : d \geq 0, \sum_{a \in \mathcal{A}} d_1(s, a) = \nu_1(s), \sum_{a \in \mathcal{A}} d_h(s, a) = \sum_{s' \in \mathcal{S}, a \in \mathcal{A}} P(s|s', a) d_{h-1}(s', a), \forall s \in S, h = 2, \ldots, H\}$. For any policy $\pi \in \Pi$, we can find a distribution $d^* \in \mathcal{D}$ such that $d^* = d^\pi$ and vice versa.

**Theorem 4.8.** *The set of feasible state-action visitation distributions $\mathcal{D}$ is convex, and there is a one-to-one correspondence between $\Pi$ and $\mathcal{D}$.*

In particular, Theorem 4.8 demonstrates that we can interpret our use of off-policy data as samples from a single policy.

**Corollary 4.9.** *If $\{d^n\}_{n=1}^N$ is a set of state-action visitation distributions of $N$ policies, define the mixture distribution as $d^{mix} = \sum_{n=1}^N \beta_n d^n$, where $\{\beta_n\}_{n=1}^N$ is a distribution of weights over these policies. Then, there exists a single policy $\pi^*$ such that sampling data from $d^{mix}$ is equivalent to sampling from the state-action visitation distribution $d^{\pi^*}$ of $\pi^*$.*

*Proof.* By the convexity of $\mathcal{D}$, the mixture distribution $d^{mix} \in \mathcal{D}$. By the one-to-one correspondence between $\mathcal{D}$ and $\Pi$, there exists a policy $\pi^* \in \Pi$ such that $d^{\pi^*} = d^{mix}$. $\square$

Corollary 4.9 tells us that during the off-policy reward updates, we can interpret our data as being generated from a single policy. Suppose we collect the same amount of new data between updates. Then, the off-policy algorithm will have $N$ times as much data compared to the on-policy algorithm to evaluate that single policy. Therefore, the off-policy algorithm can achieve better estimation error. Alternatively, to achieve a given estimation error, the off-policy algorithm has to collect less data at every iteration compared to the on-policy algorithm, which allows for more frequent updates.

In most cases, especially real world tasks, the size of the state space $S$ dominates other parameters such as the horizon length $H$ and size of the action space $A$. In these settings, the off-policy error is dominated by the estimation error. Denote the number of trajectories collected by following the current policy $B_{\text{on}}$ and $B_{\text{off}}$ in on-policy and off-policy algorithms, respectively. To achieve a similar regret, the on-policy algorithm needs to collect $N$ times as many trajectories as the off-policy algorithm (i.e., $B_{\text{on}} \approx N \cdot B_{\text{off}}$). Usually, a complete trajectory consists of thousands of interactions between the agent and the environment. Therefore, the off-policy AIL algorithm can gain significant sample efficiency.

## 5 Experiments

In addition to our theoretical analysis, we also implement our off-policy AIL algorithm with different $N$ in both discrete space MiniGrid environments (Chevalier-Boisvert et al., 2023) and continuous space OpenAI Gym MuJoCo benchmarks (Brockman et al., 2016). We compare the performance of the off-policy AIL algorithm to the on-policy version (i.e., $N = 1$). The on-policy algorithm considers the same policy updates

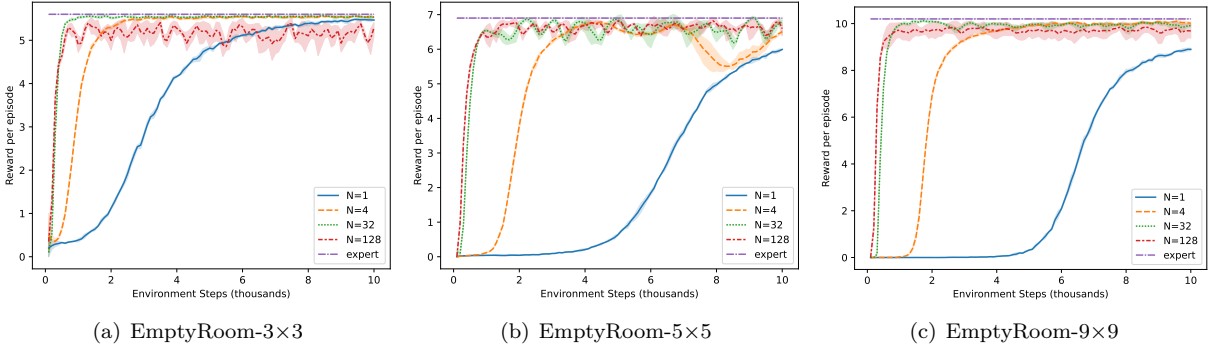

Figure 1: Experimental results for our off-policy AIL algorithm with different $N$ in three MiniGrid Empty-Room tasks with room sizes equal to $3 \times 3$, $5 \times 5$, and $9 \times 9$, respectively, from left to right. Training curves represent total reward per episode as a function of environment interactions. We evaluate the learned policy using average performance over 5 episodes. $N$ denotes the number of most recent policies we consider during reward updates, where $N = 1$ represents the on-policy algorithm. The expert's demonstration consists of 4 trajectories which are hand-crafted. We run each experiment for 5 different seeds and the shading represents the standard deviation. For more implementation details, please refer to Section C.1 in the Appendix.

as the off-policy algorithm, but only uses on-policy data for reward updates. Note that we do not compare our algorithm with state-of-the-art off-policy AIL algorithms for two reasons. First, the algorithm introduced in our work is a simplified version of popular off-policy AIL algorithms applied in practice, which allows for rigorous theoretical analysis. Second, the goal of our work is to provide theoretical support for the strong performance of these popular off-policy AIL algorithms, not to achieve state-of-the-art performance.

## 5.1 MiniGrid Environments

We begin by analyzing the behavior of the off-policy AIL algorithm in a discrete space MiniGrid environment named EmptyRoom. Within the EmptyRoom environment, the agent undertakes a navigation task with the primary objective of reaching a designated destination while minimizing the number of steps taken. Specifically, the environment is a $n \times n$ grid world with a state space $\mathcal{S} = \{(i,j)\}_{i,j=1}^{n}$, an action space $\mathcal{A} = \{\text{stay, up, down, left, right}\}$, and a horizon $H = 3n$. The agent starts from $(1, 1)$ and the destination is $(n, n)$. The agent receives a reward of 1 when in the position $(n, n)$, otherwise receives a reward of $-0.1$.

In Figure 1, we present the experimental results of the on-policy and off-policy algorithms in this task. We compare across different values of $N$, and we consider three rooms with different sizes ($n = 3, 5, 9$). Recall that $N$ is the number of most recent policies the algorithm considers when updating rewards. The experimental results show that the off-policy AIL algorithms ($N > 1$) rapidly converge to policies that align closely with the expert's performance, and demonstrate improved sample efficiency compared to the on-policy algorithm ($N = 1$). To match the expert's performance, the off-policy AIL algorithm with $N = 32$ only requires approximately 1,000 interactions between the agent and the environment in each task. The on-policy AIL algorithm, on the other hand, requires the entire training horizon of 10,000 interactions to converge in the $3 \times 3$ room, and requires additional training to converge to the expert's performance in the $5 \times 5$ and $9 \times 9$ rooms.

Among the off-policy AIL algorithms ($N > 1$), we notice that the speed of convergence suffers as the size of the room increases for small values of $N$ (i.e., $N = 4$). On the other hand, we see little benefit from increasing the value of $N$ beyond $N = 32$, which achieves strong performance in all room sizes. This observation is consistent with our theory, which tells us that there is an optimal value of $N$ that grows as the size of the state space becomes larger.

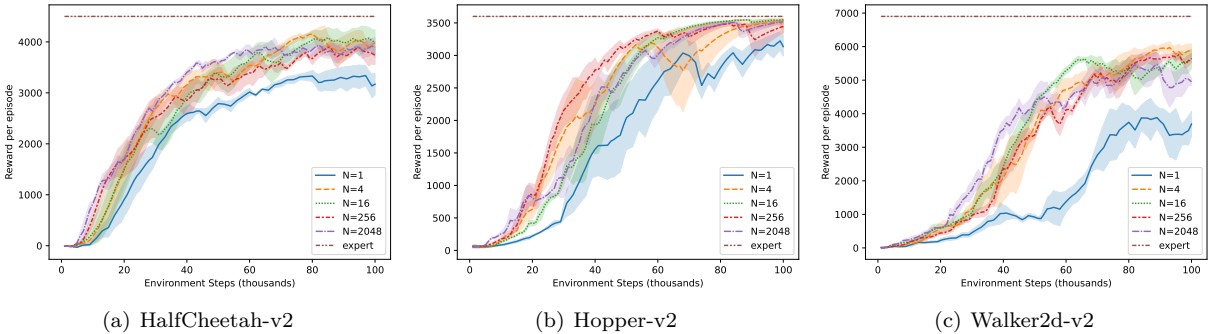

Figure 2: Experimental results for our off-policy AIL algorithm with different $N$ in three continuous space MuJoCo locomotion environments: HalfCheetah-v2, Hopper-v2, and Walker2d-v2. Training curves represent total reward per episode as a function of environment interactions. We evaluate the learned policy using average performance over 10 episodes. $N$ denotes the number of most recent policies we consider during reward updates, where $N = 1$ represents the on-policy algorithm. The expert's demonstration consists of 10 trajectories which are trained by Soft Actor-Critic (Haarnoja et al., 2018). We run each experiment for 10 different seeds and the shading represents the standard error. For more implementation details, please refer to Section C.2 in the Appendix.

## 5.2 MuJoCo Benchmarks

Additionally, we tested our algorithms in three continuous space locomotion tasks simulated in MuJoCo: HalfCheetah-v2, Hopper-v2, and Walker2d-v2. To deploy our algorithm in tasks with continuous spaces, some modifications are necessary. During policy updates, we cannot update the policy by the closed-form update rule in equation 4 which is implemented on each state. Instead, we take 5 gradient ascent steps to approximate the policy update in equation 3. Note that we consider a deep RL implementation using neural network parameterizations, so our theoretical results do not apply to this setting. Nevertheless, we are interested in analyzing whether our results provide support for the performance trends observed in a deep RL setting.

In Figure 2, we present the experimental results of the on-policy and off-policy algorithms in each continuous control task. First, comparing the on-policy algorithm ($N = 1$) and the off-policy algorithm ($N > 1$), we find that the off-policy algorithm always performs better than the on-policy algorithm for all values of $N > 1$. In every environment, the best performance is achieved by one of the off-policy AIL algorithms. Second, comparing the off-policy algorithms with different $N$, we can achieve benefits even for small values of $N$, and there is no significant benefit from increasing $N$ to very large values. Finally, the experimental results show that the optimal value of $N$ varies across environments, and there is not a very clear relationship between the environments and the optimal $N$. This may be due to the use of neural networks applied during practical training, which is not captured by our theory.

## 6 Conclusion

We studied the convergence and sample complexity of off-policy adversarial imitation learning algorithms. First, we established convergence guarantees for off-policy AIL algorithms. Furthermore, based on the regret bound we derived, we provided theoretical evidence for the sample efficiency of our off-policy algorithm. Specifically, we showed that in scenarios where the size of the state space considerably outweighs other specifications, the distribution shift error induced by off-policy updates is dominated by the benefits of having more data available. This result provides theoretical support for the benefits of off-policy AIL algorithms observed in practice. We further demonstrated the practical performance of our off-policy algorithm in discrete space MiniGrid environments and continuous space MuJoCo benchmarks. Our experimental results indicate

that the off-policy algorithm often outperforms its on-policy counterpart, while requiring substantially fewer samples.

There are several avenues for future work. First, based on our current theoretical framework, we can only prove convergence guarantees when $N = o(\sqrt{K})$. However, practically, when $N = K$, the algorithm remains effective. We believe that a sharper analysis is needed in these cases. Further, when reusing off-policy data, we simply assume a uniform distribution to construct the mixture distribution from the past policies. We conjecture that a carefully designed sampling distribution may improve the results.

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

# A  Proofs of Theorems and Lemmas in Section 4

## A.1  Proof of Lemma 4.2

*Proof.* Different from $d(s,a)$, we use $\nu(s)$ to denote the state occupancy distribution. By the definition of the state occupancy distribution, we have

$$
\begin{aligned}
||\nu_h^\pi - \nu_h^{\pi'}||_1 &= ||P_{\pi_{h-1}}\nu_{h-1}^\pi - P_{\pi'_{h-1}}\nu_{h-1}^{\pi'}||_1 \\
&= ||P_{\pi_{h-1}}\nu_{h-1}^\pi - P_{\pi'_{h-1}}\nu_{h-1}^\pi + P_{\pi'_{h-1}}\nu_{h-1}^\pi - P_{\pi'_{h-1}}\nu_{h-1}^{\pi'}||_1 \\
&\leq ||(P_{\pi_{h-1}} - P_{\pi'_{h-1}})\nu_{h-1}^\pi||_1 + ||P_{\pi'_{h-1}}(\nu_{h-1}^\pi - \nu_{h-1}^{\pi'})||_1,
\end{aligned}
\tag{6}
$$

where $P_\pi$ is the transpose of the state transition matrix induced by policy $\pi$ (i.e., $P_\pi(i,j)$ is the transition probability from state $j$ to state $i$). Following the proof of Lemma 3 in Achiam et al. (2017) for infinite discounted MDPs, we have

$$
\begin{aligned}
||(P_{\pi_{h-1}} - P_{\pi'_{h-1}})\nu_{h-1}^\pi||_1 &= \sum_{s'}\left|\sum_s (P_{\pi_{h-1}} - P_{\pi'_{h-1}})(s'|s)\nu_{h-1}^\pi(s)\right| \\
&\leq \sum_{s',s}\left|(P_{\pi_{h-1}} - P_{\pi'_{h-1}})(s'|s)\right|\nu_{h-1}^\pi(s) \\
&= \sum_{s',s}\left|\sum_a P_{h-1}(s'|s,a)(\pi_{h-1}(a|s) - \pi'_{h-1}(a|s))\right|\nu_{h-1}^\pi(s) \\
&\leq \sum_{s,a,s'} P_{h-1}(s'|s,a)\left|\pi_{h-1}(a|s) - \pi'_{h-1}(a|s)\right|\nu_{h-1}^\pi(s) \\
&= \sum_{s,a}\left|\pi_{h-1}(a|s) - \pi'_{h-1}(a|s)\right|\nu_{h-1}^\pi(s) \\
&= 2\mathbb{E}_{s\sim\nu_{h-1}^\pi}[\mathbb{D}_{\mathrm{TV}}(\pi_{h-1}||\pi'_{h-1})[s]].
\end{aligned}
\tag{7}
$$

Next, notice that

$$
||P_{\pi'_{h-1}}(\nu_{h-1}^\pi - \nu_{h-1}^{\pi'})||_1 \leq ||\nu_{h-1}^\pi - \nu_{h-1}^{\pi'}||_1.
\tag{8}
$$

By applying equation 7 and equation 8 to equation 6, we have

$$
\begin{aligned}
||\nu_h^\pi - \nu_h^{\pi'}||_1 &\leq 2\mathbb{E}_{s\sim\nu_{h-1}^\pi}[\mathbb{D}_{\mathrm{TV}}(\pi_{h-1}||\pi'_{h-1})[s]] + ||\nu_{h-1}^\pi - \nu_{h-1}^{\pi'}||_1 \\
&\leq 2\mathbb{E}_{s\sim\nu_{h-1}^\pi}[\mathbb{D}_{\mathrm{TV}}(\pi_{h-1}||\pi'_{h-1})[s]] \\
&\quad + 2\mathbb{E}_{s\sim\nu_{h-2}^\pi}[\mathbb{D}_{\mathrm{TV}}(\pi_{h-2}||\pi'_{h-2})[s]] + ||\nu_{h-2}^\pi - \nu_{h-2}^{\pi'}||_1 \\
&\;\;\vdots \\
&\leq 2\sum_{i=1}^{h-1}\mathbb{E}_{s\sim\nu_i^\pi}[\mathbb{D}_{\mathrm{TV}}(\pi_i||\pi'_i)[s]].
\end{aligned}
\tag{9}
$$

Next, we can bound the discrepancy of state-action occupancy distributions by the discrepancy of state occupancy distributions as follows:

$$
\begin{aligned}
||d_h^\pi - d_h^{\pi'}||_1 &= \sum_{s,a}\left|d_h^\pi(s,a) - d_h^{\pi'}(s,a)\right| \\
&= \sum_{s,a}\left|\nu_h^\pi(s)\pi_h(a|s) - \nu_h^\pi(s)\pi'_h(a|s) + \nu_h^\pi(s)\pi'_h(a|s) - \nu_h^{\pi'}(s)\pi'_h(a|s)\right| \\
&\leq \sum_{s,a}\nu_h^\pi(s)\left|\pi_h(a|s) - \pi'_h(a|s)\right| + \sum_s\left|\nu_h^\pi(s) - \nu_h^{\pi'}(s)\right| \\
&= 2\mathbb{E}_{s\sim\nu_h^\pi}[\mathbb{D}_{\mathrm{TV}}(\pi_h||\pi'_h)[s]] + ||\nu_h^\pi - \nu_h^{\pi'}||_1.
\end{aligned}
\tag{10}
$$

By applying equation 9 to equation 10, we have

$$||d_h^\pi - d_h^{\pi'}||_1 \le 2 \sum_{i=1}^{h} \mathbb{E}_{s \sim d_i^\pi}[\mathbb{D}_{\mathrm{TV}}(\pi_i || \pi_i')[s]].$$

□

## A.2 Proof of Theorem 4.3

*Proof.* Let $T_h(s) = \sum_a \pi_h^{k-1}(a|s) \cdot \exp\{\sigma \cdot \hat{Q}_h^{\pi^{k-1}}(s,a)\}$. Based on the policy update in equation 4, we have

$$\frac{\pi_h^k(a|s)}{\pi_h^{k-1}(a|s)} = \frac{e^{\sigma \cdot \hat{Q}_h^{\pi^{k-1}}(s,a)}}{T_h(s)}, \ \forall s \in \mathcal{S}.$$

By taking the logarithm on both sides, we see that

$$\log(\pi_h^k(a|s)) - \log(\pi_h^{k-1}(a|s)) = \sigma \cdot \hat{Q}_h^{\pi^{k-1}}(s,a) - \log(T_h(s)), \ \forall s \in \mathcal{S}.$$

The mean value theorem tells us that for $0 < p < q < 1$, there exists $y \in [p,q]$ such that

$$\frac{\log p - \log q}{p - q} = \frac{1}{y}.$$

By applying the above result, we have

$$\frac{\pi_h^k(a|s) - \pi_h^{k-1}(a|s)}{y} = \sigma \cdot \hat{Q}_h^{\pi^{k-1}}(s,a) - \log(T_h(s)), \ \forall s \in \mathcal{S}.$$

Note that $y \in (0,1)$, which implies

$$\left| \pi_h^k(a|s) - \pi_h^{k-1}(a|s) \right| \le \left| \sigma \cdot \hat{Q}_h^{\pi^{k-1}}(s,a) - \log(T_h(s)) \right|.$$

Due to the boundedness of rewards, we have $0 \le Q_h^{\pi^{k-1}}(s,a) \le H$ and $1 \le T_h(s) \le \exp\{\sigma H\}$. Together with the above inequality, we have

$$\sum_a |\pi_h^k(a|s) - \pi_h^{k-1}(a|s)| \le \sigma \sum_a |Q_h^{\pi^{k-1}}(s,a)| + \sum_a |\log T_h(s)| \le 2A\sigma H.$$

Hence, we have $\mathbb{D}_{\mathrm{TV}}(\pi_h^k || \pi_h^{k-1})[s] = \frac{1}{2} \sum_a |\pi_h^k(a|s) - \pi_h^{k-1}(a|s)| \le O(AH\sigma)$. When $\sigma = \sqrt{2 \log A/(H^2 K)}$, $\mathbb{D}_{\mathrm{TV}}(\pi_h^k || \pi_h^{k-1})[s] \le O(\sqrt{2A^2 \log A/K})$. □

## A.3 Proof of Theorem 4.4

*Proof.* This proof builds upon the proof from Liu et al. (2022b), which considered on-policy reward updates. We start from an empirical version of the reward update in equation 5 given by

$$\mu_h^{k+1} = \mathrm{Proj}_{\mu \in \mathcal{R}} \left\{ \mu_h^k + \eta \hat{\nabla}_\mu L(\pi^{k,mix}, \mu^k) \right\},$$

where we use $\hat{\nabla}$ to indicate the empirical gradient. Based on the property of projection, we have

$$(\mu_h - \mu_h^{k+1})^T (\mu_h^{k+1} - \mu_h^k - \eta \hat{\nabla}_\mu L(\pi^{k,mix}, \mu^k)) \ge 0. \tag{11}$$

By rearranging equation 11, we have

$$\eta(\mu_h - \mu_h^{k+1})^T \hat{\nabla}_{\mu_h} L(\pi^{k,mix}, \mu^k) \le (\mu_h^{k+1} - \mu_h^k)^T (\mu_h - \mu_h^{k+1})$$
$$= \frac{1}{2}(||\mu_h^k - \mu_h||_2^2 - ||\mu_h^{k+1} - \mu_h||_2^2 - ||\mu_h^{k+1} - \mu_h^k||_2^2). \tag{12}$$

Given a policy $\pi$, note that $L(\pi, \mu)$ is linear with respect to $\mu$ where $\mu = \{\mu_h\}_{h=1}^H$. Therefore, we have

$$L(\pi^k, \mu) - L(\pi^k, \mu^k) = \sum_{h=1}^H (\mu_h - \mu_h^k)^T \nabla_{\mu_h} L(\pi^k, \mu^k). \tag{13}$$

Next, by adding equation 13 to both sides of equation 12 and rearranging, we have

$$L(\pi^k, \mu) - L(\pi^k, \mu^k) \leq \sum_{h=1}^H \frac{1}{2\eta} (||\mu_h^k - \mu_h||_2^2 - ||\mu_h^{k+1} - \mu_h||_2^2 - ||\mu_h^{k+1} - \mu_h^k||_2^2)$$

$$+ \sum_{h=1}^H (\mu_h^{k+1} - \mu_h^k)^T \hat{\nabla}_{\mu_h} L(\pi^{k,mix}, \mu^k)$$

$$+ \sum_{h=1}^H (\mu_h^k - \mu_h)^T (\nabla_{\mu_h} L(\pi^{k,mix}, \mu^k) - \nabla_{\mu_h} L(\pi^k, \mu^k))$$

$$+ \sum_{h=1}^H (\mu_h^k - \mu_h)^T (\hat{\nabla}_{\mu_h} L(\pi^{k,mix}, \mu^k) - \nabla_{\mu_h} L(\pi^{k,mix}, \mu^k)).$$

Using this inequality, we can bound the regret of reward updates by

$$\text{Regret}_\mu = \sup_{\mu \in \mathcal{R}} \sum_{k=1}^K L(\pi^k, \mu) - L(\pi^k, \mu^k)$$

$$\leq \sup_{\mu \in \mathcal{R}} \sum_{k=1}^K \sum_{h=1}^H \frac{1}{2\eta} (||\mu_h^k - \mu_h||_2^2 - ||\mu_h^{k+1} - \mu_h||_2^2 - ||\mu_h^{k+1} - \mu_h^k||_2^2) \tag{14}$$

$$+ \sum_{k=1}^K \sum_{h=1}^H (\mu_h^{k+1} - \mu_h^k)^T \hat{\nabla}_{\mu_h} L(\pi^{k,mix}, \mu^k) \tag{15}$$

$$+ \sup_{\mu \in \mathcal{R}} \sum_{k=1}^K \sum_{h=1}^H (\mu_h^k - \mu_h)^T (\nabla_{\mu_h} L(\pi^{k,mix}, \mu^k) - \nabla_{\mu_h} L(\pi^k, \mu^k)) \tag{16}$$

$$+ \sup_{\mu \in \mathcal{R}} \sum_{k=1}^K \sum_{h=1}^H (\mu_h^k - \mu_h)^T (\hat{\nabla}_{\mu_h} L(\pi^{k,mix}, \mu^k) - \nabla_{\mu_h} L(\pi^{k,mix}, \mu^k)). \tag{17}$$

Next, we bound each of these terms individually. We can bound equation 14 by

$$\sup_{\mu \in \mathcal{R}} \sum_{k=1}^K \sum_{h=1}^H \frac{1}{2\eta} (||\mu_h^k - \mu_h||_2^2 - ||\mu_h^{k+1} - \mu_h||_2^2 - ||\mu_h^{k+1} - \mu_h^k||_2^2)$$

$$\leq \frac{1}{2\eta} \sup_{\mu \in \mathcal{R}} \sum_{h=1}^H (||\mu_h^1 - \mu_h||_2^2 - ||\mu_h^{K+1} - \mu_h||_2^2) \leq \frac{HSA}{2\eta}. \tag{18}$$

The last inequality holds because $\mu_h(s, a) \in [0, 1], \forall s \in \mathcal{S}, a \in \mathcal{A}, h \in [H]$.

We can bound equation 15 by

$$\sum_{k=1}^K \sum_{h=1}^H (\mu_h^{k+1} - \mu_h^k)^T \hat{\nabla}_{\mu_h} L(\pi^{k,mix}, \mu^k) \leq \sum_{k=1}^K \sum_{h=1}^H ||\mu_h^{k+1} - \mu_h^k||_2 \cdot ||\hat{\nabla}_{\mu_h} L(\pi^{k,mix}, \mu^k)||_2$$

$$\leq \eta \sum_{k=1}^K \sum_{h=1}^H ||\hat{\nabla}_{\mu_h} L(\pi^{k,mix}, \mu^k)||_2^2$$

$$\leq 4\eta HK. \tag{19}$$

The first inequality holds by Cauchy-Schwarz inequality. The second inequality holds due to $||\mu_h^{k+1} - \mu_h^k||_2 \leq ||\eta \hat{\nabla}_{\mu_h} L(\pi^{mix}, \mu^k)||_2$ based on equation 5 and the property of projection. The last inequality holds because $||\hat{\nabla}_{\mu_h} L(\pi^{k,mix}, \mu^k)||_2 = ||\hat{d}_h^{\pi^E} - \hat{d}_h^{\pi^{k,mix}}||_2 \leq 2$.

In order to bound equation 16, we first apply Hölder's inequality to see that

$$\sup_{\mu \in \mathcal{R}} \sum_{k=1}^{K} \sum_{h=1}^{H} (\mu_h^k - \mu_h)^T (\nabla_{\mu_h} L(\pi^{k,mix}, \mu^k) - \nabla_{\mu_h} L(\pi^k, \mu^k))$$

$$\leq \sup_{\mu \in \mathcal{R}} \sum_{k=1}^{K} \sum_{h=1}^{H} ||\mu_h^k - \mu_h||_\infty \cdot ||\nabla_{\mu_h} L(\pi^{k,mix}, \mu^k) - \nabla_{\mu_h} L(\pi^k, \mu^k))||_1. \quad (20)$$

By definition, we have

$$\nabla_{\mu_h} L(\pi^{k,mix}, \mu^k) - \nabla_{\mu_h} L(\pi^k, \mu^k) = \nabla_{\mu_h} J(\pi^k, \mu^k) - \nabla_{\mu_h} J(\pi^{k,mix}, \mu^k)$$

$$= d_h^{\pi^k} - \sum_{n=1}^{N(k)} \beta_n d_h^{\pi^{k+1-n}}. \quad (21)$$

We assume the sampling weights of each policy are the same (i.e., $\beta_1 = \cdots = \beta_{N(k)} = \frac{1}{N(k)}$). Then,

$$d_h^{\pi^k} - \sum_{n=1}^{N(k)} \beta_n d_h^{\pi^{k+1-n}} = \sum_{n=1}^{N(k)} \beta_n (d_h^{\pi^k} - d_h^{\pi^{k+1-n}})$$

$$= \frac{1}{N(k)} \sum_{n=1}^{N(k)} (d_h^{\pi^k} - d_h^{\pi^{k+1-n}})$$

$$= \frac{1}{N(k)} \sum_{n=1}^{N(k)-1} (N(k) - n) \cdot (d_h^{\pi^{k+1-n}} - d_h^{\pi^{k-n}}). \quad (22)$$

Based on Lemma 4.2 and Theorem 4.3, we have that

$$||d_h^{\pi^{k+1-n}} - d_h^{\pi^{k-n}}||_1 \leq 2 \sum_{i=1}^{h} \mathbb{E}_{s \sim d_i^{\pi^{k-n}}} [\mathbb{D}_{\text{TV}}(\pi_i^{k+1-n} || \pi_i^{k-n})[s]] \leq \tilde{O}\left(\frac{Ah}{\sqrt{K}}\right). \quad (23)$$

By applying equation 22 and equation 23 to equation 21, we have

$$||\nabla_{\mu_h} L(\pi^{k,mix}, \mu^k) - \nabla_{\mu_h} L(\pi^k, \mu^k))||_1 \leq \frac{1}{N(k)} \sum_{n=1}^{N(k)-1} (N(k) - n) ||d_h^{\pi^{k+1-n}} - d_h^{\pi^{k-n}}||_1$$

$$\leq \tilde{O}\left(\frac{Ah(N(k) - 1)}{\sqrt{K}}\right). \quad (24)$$

Then, we apply equation 24 to equation 20 and use the fact that $\mu_h(s,a) \in [0,1]$ to show

$$\sup_{\mu \in \mathcal{R}} \sum_{k=1}^{K} \sum_{h=1}^{H} (\mu_h^k - \mu_h)^T (\nabla_{\mu_h} L(\pi^{k,mix}, \mu^k) - \nabla_{\mu_h} L(\pi^k, \mu^k))$$

$$\leq \sup_{\mu \in \mathcal{R}} \sum_{k=1}^{K} \sum_{h=1}^{H} ||\mu_h^k - \mu_h||_\infty \cdot ||\nabla_{\mu_h} L(\pi^{k,mix}, \mu^k) - \nabla_{\mu_h} L(\pi^k, \mu^k))||_1$$

$$\leq \tilde{O}\left(\sqrt{\frac{H^4 A^2 (\sum_k (N(k) - 1))^2}{K}}\right). \quad (25)$$

In order to bound equation 17, note that $\pi^{k,mix}$ can be regarded as a single policy based on Corollary 4.9. Therefore, we can bound equation 17 in the same way as in Liu et al. (2022b) (Lemma H.8).

**Lemma A.1** (Lemma H.8 in Liu et al. 2022b for tabular MDPs). *With probability at least $1 - \delta$, it holds that*

$$\sup_{\mu \in \mathcal{R}} \sum_{k=1}^{K} \sum_{h=1}^{H} (\mu_h^k - \mu_h)^T (\hat{\nabla}_{\mu_h} L(\pi^{k,mix}, \mu^k) - \nabla_{\mu_h} L(\pi^{k,mix}, \mu^k)) \leq \tilde{O}(\sqrt{H^3 SAK}),$$

*where $\hat{\nabla}_{\mu_h} L(\pi^{k,mix}, \mu^k)$ is estimated with one sample.*

By applying this result and using high-probability Azuma-Hoeffding inequality, with probability at least $1 - \delta$ we have that

$$\sup_{\mu \in \mathcal{R}} \sum_{k=1}^{K} \sum_{h=1}^{H} (\mu_h^k - \mu_h)^T (\hat{\nabla}_{\mu_h} L(\pi^{k,mix}, \mu^k) - \nabla_{\mu_h} L(\pi^{k,mix}, \mu^k))$$

$$= \sup_{\mu \in \mathcal{R}} \sum_{k=1}^{K} \sum_{h=1}^{H} (\mu_h^k - \mu_h)^T \left( d_h^{\pi^{k,mix}} - \hat{d}_h^{\pi^{k,mix}} \right)$$

$$\leq \tilde{O} \left( \sqrt{\frac{H^3 SAK^2}{B \sum_k N(k)}} \right), \tag{26}$$

where $BN(k)$ is number of samples used to estimate $\hat{\nabla}_{\mu_h} L(\pi^{k,mix}, \mu^k)$.

Recall that we assume the number of expert trajectories is infinite, so we do not need to consider the estimation error related to the expert's policy. This assumption does not affect our theoretical results as it is only related to the number of available trajectories of the expert policy, instead of on-policy or off-policy data from behavior policies.

By combining the results from equation 18, equation 19, equation 25, and equation 26 and setting $\eta = \sqrt{\frac{SA}{K}}$, with probability at least $1 - \delta$ we have

$$\sup_{\mu \in \mathcal{R}} \sum_{k=1}^{K} L(\pi^k, \mu) - L(\pi^k, \mu^k)$$

$$\leq \tilde{O} \left( \frac{HSA}{\eta} + \eta HK + \sqrt{\frac{H^4 A^2 (\sum_k (N(k) - 1))^2}{K}} + \sqrt{\frac{H^3 SAK^2}{B \sum_k N(k)}} \right)$$

$$= \tilde{O} \left( \sqrt{H^2 SAK} + \sqrt{\frac{H^4 A^2 (\sum_k (N(k) - 1))^2}{K}} + \sqrt{\frac{H^3 SAK^2}{B \sum_k N(k)}} \right).$$

Specifically, if $N$ is fixed, then with probability at least $1 - \delta$ we have

$$\sup_{\mu \in \mathcal{R}} \sum_{k=1}^{K} L(\pi^k, \mu) - L(\pi^k, \mu^k) \leq \tilde{O} \left( \frac{HSA}{\eta} + \eta HK + \sqrt{H^4 (N-1)^2 A^2 K} + \sqrt{\frac{H^3 SAK}{BN}} \right)$$

$$= \tilde{O} \left( \sqrt{H^2 SAK} + \sqrt{H^4 (N-1)^2 A^2 K} + \sqrt{\frac{H^3 SAK}{BN}} \right).$$

$\square$

## A.4   Proof of Theorem 4.8

*Proof.* First, the convexity of $\mathcal{D}$ can be easily verified because all constraints are linear. Second, without loss of generality, we can assume $d(s) > 0$, $\forall s \in \mathcal{S}$. Otherwise, if there exists $s \in \mathcal{S}$ such that $d(s) = 0$, we

can take arbitrary actions at this state and it is enough to consider the state space $\mathcal{S} \setminus \{s\}$. Based on this assumption, we define

$$\pi_h(a|s) \triangleq d_h(s,a)/\sum_{a'} d_h(s,a'), \ \forall s \in \mathcal{S}.$$

It can be verified that policy $\pi = \{\pi_h\}_{h=1}^H$ and state-action visitation distribution $d = \{d_h\}_{h=1}^H$ has a one-to-one correspondence. The proof is extended from Syed et al. (2008), where the setting was infinite discounted MDPs. First, given a policy $\pi$, it is trivial to show that its occupancy distribution $d^\pi = [d_1, \ldots, d_H]$ is in the set $\mathcal{D}$. Next we will show, given a feasible distribution $d \in \mathcal{D}$, it can be verified that $\pi_h(a|s)$ defined above is the unique policy that has the same occupancy distribution (we have assumed $d(s) > 0, \forall s \in \mathcal{S}, \ h \in [H]$). First, by the definition $d_1^\pi(s,a) = (\sum_{a'} d_1(s,a')) \pi_1(a|s)$, $\pi_1(a|s)$ is uniquely defined by

$$\pi_1(a|s) = \frac{d_1^\pi(s,a)}{\sum_{a'} d_1^\pi(s,a')}.$$

For $h = 2, \ldots, H$, we have

$$d_h^\pi(s,a) = \pi_h(a|s) \sum_{s',a'} d_{h-1}^\pi(s',a')P(s|s',a'),$$

so it follows that

$$\pi_h(a|s) = \frac{d_h^\pi(s,a)}{\sum_{s',a'} d_{h-1}^\pi(s',a')P(s|s',a')}.$$

Finally, note that $\sum_{s',a'} d_{h-1}^\pi(s',a')P(s|s',a') = \sum_a d_h^\pi(s,a)$. Therefore, we have

$$\pi_h(a|s) = \frac{d_h^\pi(s,a)}{\sum_{a'} d_h^\pi(s,a')}.$$

$\qquad\qquad\qquad\qquad\qquad\qquad\qquad\qquad\qquad\qquad\qquad\qquad\qquad\qquad\qquad\qquad\qquad\quad \square$

## B  Extension to linear MDPs

In this section, we extend our results to linear MDPs. Our analysis builds upon the framework in Liu et al. (2022b), which considered on-policy reward updates in linear MDPs.

**Definition B.1** (Linear MDPs). We define linear MDPs as in Liu et al. (2022b); Jin et al. (2020); Agarwal et al. (2020). A MDP is linear if its transition function and reward function are linear based on known feature spaces, i.e., there exists a feature map $\psi : \mathcal{S} \times \mathcal{A} \times \mathcal{S} \to \mathbb{R}^d$ and $\theta_h \in \mathbb{R}^d$ such that

$$P_h(s'|s,a) = \psi(s,a,s')^T \theta_h, \ \forall (s,a,s') \in \mathcal{S} \times \mathcal{A} \times \mathcal{S},$$

and there exists a feature map $\phi : \mathcal{S} \times \mathcal{A} \to \mathbb{R}^d$ and $\mu_h \in \mathbb{R}^d$ such that

$$r_h(s,a) = \phi(s,a)^T \mu_h, \forall (s,a) \in \mathcal{S} \times \mathcal{A}.$$

**Assumption B.2.** We assume that the state space $\mathcal{S}$ and action space $\mathcal{A}$ are measurable sets with finite measures $S$ and $A$, respectively.

**Assumption B.3.** We assume that $||\theta_h||_2 \leq \sqrt{d}$ and there exists an absolute constant $R > 0$ such that

$$R^{-2} \cdot \sup_{s' \in \mathcal{S}} |\psi(s,a,s')^T y|^2 \leq \int_{s' \in \mathcal{S}} |\psi(s,a,s')^T y|^2 ds' \leq d,$$

for any $(s,a) \in \mathcal{S} \times \mathcal{A}$ and $y \in \mathbb{R}^d$ with $||y||_2 \leq 1$.

**Assumption B.4.** We assume that $||\mu_h||_2 \leq \sqrt{d}$ and $||\phi(s,a)||_2 \leq 1, \forall (s,a) \in \mathcal{S} \times \mathcal{A}$ which ensures that $|r_h^\mu(s,a)| \leq \sqrt{d}$.

**Lemma B.5** (Lemma 4.1 in linear MDPs). *Let $\sigma = \sqrt{\frac{2 \log A}{H^2 \sqrt{d} K}}$. With probability at least $1 - \delta$, the regret of policy updates in linear MDPs is bounded by*

$$Regret_\pi \leq \tilde{O}(\sqrt{H^4 K d^3}).$$

*Proof.* Refer to Appendix H in Liu et al. (2022b). $\qquad\square$

**Lemma B.6** (Lemma 4.2 in linear MDPs). *In finite-horizon undiscounted MDPs, the divergence between state-action visitation distributions is bounded by the total variation distance of the corresponding policies according to*

$$||d_h^\pi - d_h^{\pi'}||_1 = \int_{s,a} \left| d_h^\pi - d_h^{\pi'} \right| ds da \leq 2 \sum_{i=1}^{h} \mathbb{E}_{s \sim \nu_i^\pi}[\mathbb{D}_{\mathrm{TV}}(\pi_i || \pi_i')[s]].$$

*Proof.* It can be proved by substituting $\sum_{s,a}$ with $\int_{s,a \in \mathcal{S} \times \mathcal{A}}$ and regarding $P(s'|s,a)$ as a probability density function in the proof of Lemma 4.2. $\qquad\square$

**Theorem B.7** (Theorem 4.3 in linear MDPs). *The total variation distance between two consecutive policies induced by implementing the policy update algorithm in equation 3 is bounded by*

$$\mathbb{D}_{\mathrm{TV}}(\pi_h^k || \pi_h^{k-1})[s] \leq O(AH\sigma\sqrt{d}), \ \forall s \in \mathcal{S}, k \in [K], h \in [H].$$

*When $\sigma = \sqrt{\frac{2 \log A}{H^2 \sqrt{d} K}}$, the bound becomes $\tilde{O}(\sqrt{\frac{A^2 \sqrt{d}}{K}})$.*

*Proof.* Let $T_h(s) = \int_{a \in \mathcal{A}} \pi_h^{k-1}(a|s) \cdot \exp\{\sigma \cdot \hat{Q}_h^{\pi^{k-1}}(s,a)\} da$. Based on the policy update in equation 4, we have

$$\frac{\pi_h^k(a|s)}{\pi_h^{k-1}(a|s)} = \frac{e^{\sigma \cdot \hat{Q}_h^{\pi^{k-1}}(s,a)}}{T_h(s)}, \ \forall s \in \mathcal{S}.$$

Then, as in the proof of Theorem 4.3, we have

$$\left| \pi_h^k(a|s) - \pi_h^{k-1}(a|s) \right| \leq \left| \sigma \cdot \hat{Q}_h^{\pi^{k-1}}(s,a) - \log(T_h(s)) \right|.$$

Due to the boundedness of rewards, we have $0 \leq Q_h^{\pi^{k-1}}(s,a) \leq H\sqrt{d}$ and $1 \leq T_h(s) \leq \exp\{\sigma H \sqrt{d}\}$. Together with the above inequality, we have

$$\int_{a \in \mathcal{A}} |\pi_h^k(a|s) - \pi_h^{k-1}(a|s)| da \leq \sigma \int_{a \in \mathcal{A}} |Q_h^{\pi^{k-1}}(s,a)| da + \int_{a \in \mathcal{A}} |\log T_h(s)| da \leq 2A\sigma H\sqrt{d}.$$

Hence, we have $\mathbb{D}_{\mathrm{TV}}(\pi_h^k || \pi_h^{k-1})[s] = \frac{1}{2} \int_{a \in \mathcal{A}} |\pi_h^k(a|s) - \pi_h^{k-1}(a|s)| da \leq O(AH\sigma\sqrt{d})$. When $\sigma = \sqrt{\frac{2 \log A}{H^2 \sqrt{d} K}}$, $\mathbb{D}_{\mathrm{TV}}(\pi_h^k || \pi_h^{k-1})[s] \leq \tilde{O}(\sqrt{\frac{A^2 \sqrt{d}}{K}})$. $\qquad\square$

**Theorem B.8** (Theorem 4.4 in linear MDPs). *Let $\eta = \sqrt{\frac{d}{K}}$ and $\sigma = \sqrt{\frac{2 \log A}{H^2 \sqrt{d} K}}$. Then, with probability at least $1 - \delta$, the regret of the reward update problem is bounded by*

$$Regret_\mu = \sup_{\mu \in \mathcal{R}} \sum_{k=1}^{K} L(\pi^k, \mu) - L(\pi^k, \mu^k)$$

$$\leq \tilde{O}(\underbrace{\sqrt{H^2 dK}}_{regret\ of\ reward\ updates} + \underbrace{\sqrt{\frac{H^4 A^2 (\sum_k N(k) - 1)^2 d^{3/2}}{K}}}_{error\ induced\ by\ off\text{-}policy\ updates} + \underbrace{\sqrt{\frac{H^3 d^2 K^2}{B \sum_k N(k)}}}_{estimation\ error}),$$

*where $B$ is the number of new trajectories collected with the current policy between updates. Specifically, when $N$ is fixed, we have*

$$Regret_\mu \leq \tilde{O}\big( \underbrace{\sqrt{H^2 dK}}_{regret\ of\ reward\ updates} + \underbrace{\sqrt{H^4 A^2 (N-1)^2 K d^{3/2}}}_{error\ induced\ by\ off\text{-}policy\ updates} + \underbrace{\sqrt{\frac{H^3 d^2 K}{BN}}}_{estimation\ error} \big).$$

*Proof.* In linear MDPs, we can bound the regret of reward updates in the same way as the tabular setting. As in the proof of Theorem 4.4, we have

$$Regret_\mu = \sup_{\mu \in \mathcal{R}} \sum_{k=1}^{K} L(\pi^k, \mu) - L(\pi^k, \mu^k)$$

$$\leq \sup_{\mu \in \mathcal{R}} \sum_{k=1}^{K} \sum_{h=1}^{H} \frac{1}{2\eta}(||\mu_h^k - \mu_h||_2^2 - ||\mu_h^{k+1} - \mu_h||_2^2 - ||\mu_h^{k+1} - \mu_h^k||_2^2) \tag{27}$$

$$+ \sum_{k=1}^{K} \sum_{h=1}^{H} (\mu_h^{k+1} - \mu_h^k)^T \hat{\nabla}_{\mu_h} L(\pi^{k,mix}, \mu^k) \tag{28}$$

$$+ \sup_{\mu \in \mathcal{R}} \sum_{k=1}^{K} \sum_{h=1}^{H} (\mu_h^k - \mu_h)^T (\nabla_{\mu_h} L(\pi^{k,mix}, \mu^k) - \nabla_{\mu_h} L(\pi^k, \mu^k)) \tag{29}$$

$$+ \sup_{\mu \in \mathcal{R}} \sum_{k=1}^{K} \sum_{h=1}^{H} (\mu_h^k - \mu_h)^T (\hat{\nabla}_{\mu_h} L(\pi^{k,mix}, \mu^k) - \nabla_{\mu_h} L(\pi^{k,mix}, \mu^k)). \tag{30}$$

Next, we bound each of these terms individually. We can bound equation 27 by

$$\sup_{\mu \in \mathcal{R}} \sum_{k=1}^{K} \sum_{h=1}^{H} \frac{1}{2\eta}(||\mu_h^k - \mu_h||_2^2 - ||\mu_h^{k+1} - \mu_h||_2^2 - ||\mu_h^{k+1} - \mu_h^k||_2^2) \leq \sup_{\mu \in \mathcal{R}} \frac{1}{2\eta} \sum_{h=1}^{H} ||\mu_h^1 - \mu||_2^2 \leq \frac{2}{\eta} Hd,$$

where the last inequality holds due to the assumption $||\mu_h||_2 \leq \sqrt{d}$.

We can bound equation 28 by

$$\sum_{k=1}^{K} \sum_{h=1}^{H} (\mu_h^{k+1} - \mu_h^k)^T \hat{\nabla}_{\mu_h} L(\pi^{k,mix}, \mu^k) \leq \sum_{k=1}^{K} \sum_{h=1}^{H} \eta ||\hat{\nabla}_{\mu_h} L(\pi^{k,mix}, \mu^k)||_2^2 \leq 4\eta HK,$$

where the first inequality was shown in the proof of Theorem 4.4 and the second inequality holds because $||\hat{\nabla}_{\mu_h} L(\pi^{k,mix}, \mu^k)||_2 \leq 2||\phi(\cdot, \cdot)||_2 \leq 2$.

In order to bound equation 29, note that

$$J(\pi, \mu) = \sum_{h=1}^{H} \int_{s,a} d_h^\pi(s, a) \phi(s, a)^T \mu_h ds da.$$

By definition, we have

$$\nabla_{\mu_h} L(\pi^{k,mix}, \mu^k) - \nabla_{\mu_h} L(\pi^k, \mu^k) = \int_{s,a} \left( d_h^{\pi^k}(s, a) - \frac{1}{N(k)} \sum_{n=1}^{N(k)} d_h^{\pi^{k+1-n}}(s, a) \right) \phi(s, a) ds da.$$

Based on Theorem 4.3 and Lemma 4.2 in linear MDPs, we have

$$\sup_{\mu \in \mathcal{R}} \sum_{k=1}^{K} \sum_{h=1}^{H} (\mu_h^k - \mu_h)^T (\nabla_{\mu_h} L(\pi^{k,mix}, \mu^k) - \nabla_{\mu_h} L(\pi^k, \mu^k))$$

$$\leq \sup_{\mu \in \mathcal{R}} \sum_{k=1}^{K} \sum_{h=1}^{H} \sup_{s,a} |\phi(s,a)^T (\mu_h^k - \mu_h)| \cdot \| \frac{1}{N(k)} \sum_{n=1}^{N(k)} (d_h^{\pi^k} - d_h^{\pi^{k+1-n}}) \|_1$$

$$\leq \sup_{\mu \in \mathcal{R}} \sum_{k=1}^{K} \sum_{h=1}^{H} 2\sqrt{d} \| \frac{1}{N(k)} \sum_{n=1}^{N(k)} (d_h^{\pi^k} - d_h^{\pi^{k+1-n}}) \|_1$$

$$\leq \tilde{O}(\sqrt{\frac{H^4 A^2 (\sum_k N(k) - 1)^2 d^{3/2}}{K}}),$$

where the last inequality follows from

$$\| \frac{1}{N(k)} \sum_{n=1}^{N(k)} (d_h^{\pi^k} - d_h^{\pi^{k+1-n}}) \|_1 \leq \frac{1}{N(k)} \sum_{n=1}^{N(k)-1} (N(k) - n) \cdot \| d_h^{\pi^{k+1-n}} - d_h^{\pi^{k-n}} \|_1$$

$$\leq \tilde{O}\left( h(N(k) - 1) \sqrt{\frac{A^2 \sqrt{d}}{K}} \right).$$

Same as the tabular case, in order to bound equation 30, note that $\pi^{k,mix}$ can be regarded as a single policy based on Corollary 4.9. Therefore, we can bound equation 30 in the same way as in Liu et al. (2022b) (Lemma H.8) with high-probability Azuma-Hoeffding inequality. By applying this result, with probability at least $1 - \delta$ we have that

$$\sup_{\mu \in \mathcal{R}} \sum_{k=1}^{K} \sum_{h=1}^{H} (\mu_h^k - \mu_h)^T (\hat{\nabla}_{\mu_h} L(\pi^{k,mix}, \mu^k) - \nabla_{\mu_h} L(\pi^{k,mix}, \mu^k))$$

$$= \sup_{\mu \in \mathcal{R}} \sum_{k=1}^{K} \sum_{h=1}^{H} (\mu_h^k - \mu_h)^T \int_{s,a} \left( d_h^{\pi^{k,mix}}(s,a) - \hat{d}_h^{\pi^{k,mix}}(s,a) \right) \phi(s,a) ds da$$

$$\leq \tilde{O}\left( \sqrt{\frac{H^3 d^2 K^2}{B \sum_k N(k)}} \right).$$

By combining the upper bounds of equation 27, equation 28, equation 29, and equation 30 and setting $\eta = \sqrt{\frac{d}{K}}$, with probability at least $1 - \delta$ the regret of reward updates in linear MDPs is bounded by

$$Regret_\mu = \sup_{\mu \in \mathcal{R}} \sum_{k=1}^{K} L(\pi^k, \mu) - L(\pi^k, \mu^k)$$

$$\leq \tilde{O}( \underbrace{\sqrt{H^2 dK}}_{regret\ of\ reward\ updates} + \underbrace{\sqrt{\frac{H^4 A^2 (\sum_k N(k) - 1)^2 d^{3/2}}{K}}}_{error\ induced\ by\ off\text{-}policy\ updates} + \underbrace{\sqrt{\frac{H^3 d^2 K^2}{B \sum_k N(K)}}}_{estimation\ error} ).$$

Specifically, when $N$ is fixed, we have

$$Regret_\mu \leq \tilde{O}( \underbrace{\sqrt{H^2 dK}}_{regret\ of\ reward\ updates} + \underbrace{\sqrt{H^4 A^2 (N-1)^2 K d^{3/2}}}_{error\ induced\ by\ off\text{-}policy\ updates} + \underbrace{\sqrt{\frac{H^3 d^2 K}{BN}}}_{estimation\ error} ).$$

$\square$

## C   Implementation Details

### C.1   MiniGrid Environments

For MiniGrid EmptyRoom tasks, we exactly follow the algorithm introduced in the main paper which consists of policy updates given by equation 4 and reward updates given by equation 5. The only difference is that we do not learn the transition function for policy updates. Instead, we assume the transition function is known when we estimate Q-values, since it only impacts the regret of policy updates which is not the focus of this work. We make a small modification to the MiniGrid action set by changing it from {stay, go forward, turn left, turn right} to {stay, up, down, left, right}.

We consider the learning rates $\sigma = 10\sqrt{\frac{2\log(4)}{H^2 K}}$ and $\eta = \frac{5}{\sqrt{K}}$ for policy updates and reward updates, respectively. We train each algorithm for 10,000 environment interactions. We set the horizon $H$ of an episode to $3n$, where $n \times n$ is the size of the room. We maintain a replay buffer with size of 128, and we sample 32 data points from the replay buffer when we conduct reward updates. The experiments were run with two A5000 GPUs (24G memory) and it took approximately 5 minutes for each environment and each seed.

### C.2   MuJoCo Benchmarks

For continuous space MuJoCo locomotion tasks, we need to make some modifications to our algorithm. Specifically, for policy updates, we use off-policy mirror descent policy optimization (MDPO) algorithm introduced by Tomar et al. (2021). With this modification and by taking multiple gradient steps (which we set as 5 per iteration), we achieve a deep variant of the policy update in equation 3. We parameterize the actor and critic with neural networks, using an MLP architecture with 2 hidden layers and 256 hidden units. We use Adam optimizer with learning rate $10^{-3}$, $10^{-5}$ for actor and critic, respectively. The regularization coefficient we use is 0.1.

For reward updates, we use the discriminator introduced in Ho & Ermon (2016) with the modification that the data comes from multiple past policies. Although this is a different loss compared to the loss we use for theoretical analysis, the only difference is an additional regularization which can be regarded as a technique for practical performance. The learning rate of the discriminator is $10^{-3}$. We train each algorithm for 100,000 environment interactions. We maintain a replay buffer with size of 2,048, and we sample 256 data points from the replay buffer when we conduct reward updates. The experiments were run with two A5000 GPUs (24G memory) and it took approximately 5 hours for each environment and each seed.

# D    Additional Experimental Results

In Figure 1, we showed results using model-based policy updates. Here, we explore an alternative algorithm family based on policy gradient updates on the MiniGrid environment. The results are presented in Figure 3.

Across the EmptyRoom environments with all sizes, the off-policy versions ($N > 1$) exhibit faster convergence and superior final performance compared to the on-policy version, consistent with our theoretical findings. Besides, in EmptyRoom-$5 \times 5$, $N = 32$ outperforms $N = 128$, whereas in EmptyRoom-$9 \times 9$, $N = 128$ achieves nearly the same performance as $N = 32$. This observation also aligns with our theoretical results: as the state space of the environment grows, the optimal number of recent policies to use for reward updates shifts upward.

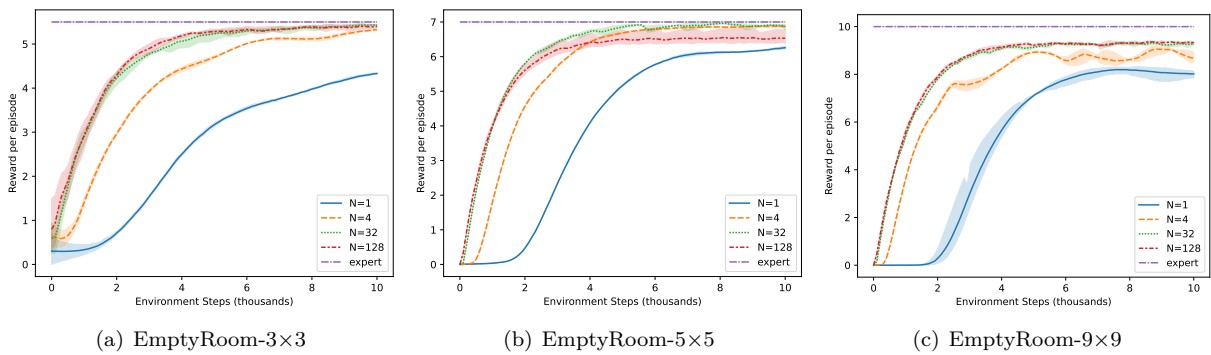

(a) EmptyRoom-3×3          (b) EmptyRoom-5×5          (c) EmptyRoom-9×9

Figure 3: Experimental results for off-policy AIL algorithm with policy-gradient updates for different $N$ in three MiniGrid EmptyRoom tasks with room sizes equal to $3 \times 3$, $5 \times 5$, and $9 \times 9$, respectively, from left to right. Training curves represent total reward per episode as a function of environment interactions. We evaluate the learned policy using average performance over 5 episodes. $N$ denotes the number of most recent policies we consider during reward updates, where $N = 1$ represents the on-policy algorithm. The expert's demonstration consists of 4 trajectories which are hand-crafted. We run each experiment for 5 different seeds and the shading represents the standard deviation.

