# OpenReview forum: "Provably Efficient Off-Policy Adversarial Imitation Learning with Convergence Guarantees"
_TMLR — Accepted by TMLR_

### Review · Reviewer_Uipr · 2025-12-11

**Summary Of Contributions:**

The paper studies off-policy adversarial imitation learning, and shows that a alternating gradient descent algorithm that uses off-policy samples generated by the past $N$ policy iterates achieves sublinear regret. A nice feature of the algorithm is that it does not have to perform importance sampling correction and simply treats off-policy samples as if they are on-policy. This makes the implementation of this algorithm convenient in practice. The regret bound shows explicit dependencies on important structural parameters including $S,A,H$, revealing the conditions under which it is favorable to use $N>1$ (according to the derived upper bound). Numerical simulations verifies the theoretical results -- it is observed that as $N$ becomes large, the policy improvement is initially fast but eventually suffers an asymptotic bias, whereas if $N$ is too small, the policy improvement slows down.

**Audience:**

Yes

**Audience Explanation:**

The work is a technically solid work in RL theory, and may be of interest to researchers of this community.

**Broader Impact Concerns:**

No concerns

**Claims And Evidence:**

Yes

**Claims Explanation:**

The paper's theoretical results are supported by proofs. I did not check the proof details but I find the claims in all theorems, propositions, and lemmas credible and sound. The authors also conduct small-scale simulations to verify the theory. The introduction, development, and presentation of the theoretical results are clear and well-written.

**Requested Changes:**

1) I would like to note that the technical innovation of this work is quite limited. The regret bound associated with the policy updates is from Shani et al. (2022). To derive the regret bound for reward updates, an important step is to show that the distance between state visitation distributions is bounded by the distance between the policies that induce the state visitation. This has been a very standard and well-known result. The authors state that prior literature only showed this result in infinite-horizon discounted-reward MDP, but I do not see any technical challenges in deriving the same result in the finite-horizon total-reward setting. The theoretical results in Section 4.2 are also very standard and straightforward to show.

That said, I understand that TMLR puts less emphasis on novelty and instead focuses on technical correctness. This paper reveals some interesting insight in off-policy AIL and is technically correct and sound to my knowledge.

2) > In order to ensure sample efficiency for the RL problem, we will apply a model-based method used in previous works (Shani et al., 2022; Liu et al., 2022b).

I understand that the authors probably considered this KL-regularized model-based RL algorithm to plug in the regret bound from Shani et al. (2022). But for clarity and potential broader applicability, the authors should clarify if the policy update is necessarily restricted to this particular algorithm.

3) Regarding Eq. (5), it is not clear to me how we can compute this gradient deterministically in practical problems. Perhaps we can get some reliable stochastic estimates of the gradient using samples, but stochastic noise will likely affect the regret bound.

4) What does estimation error mean in Theorem 4.4?

5) What is the best choice of $N(k)$ that optimizes the upper bound in Theorem 4.4 if it is not restricted to be a constant?

6) The authors separately establish policy regret and reward regret but do not combine them into an overall AIL regret bound. I wonder what prevents the authors from carrying out this final, seemingly straightforward step, and what is the choice of $N$ that optimizes this overall regret bound.

---

> ### Author Response · Authors · 2026-02-12
>
> We appreciate the reviewer's valuable comments and positive feedbacks on our work. We have carefully addressed your questions below and made corresponding modifications to the manuscripts. We hope the updated version addresses your concerns.
>
> > 1. The technical innovation of this work is quite limited.
>
> We thank the reviewer mentioned TMLR puts less emphasis on novelty and instead focuses on technical correctness and admitted that our work reveals some interesting insight in off-policy AIL. We agree that our work mainly builds on the existing AIL frameworks, but we also want to emphasize that our work successfully bridges the gap between the theory and practice of AIL by showing the convergence of off-policy AIL which is an interesting and meaningful result.
>
> > 2. More algorithm options for policy updates
>
> We have updated the manuscript to discuss a broader range of algorithms for policy updates. In short words, the policy updates alone is simply an adversarial MDP problem in the episodic setting with unknown transitions and full information. Any algorithms that can achieve sub-linear regret in this setting can be applied. However, when considering off-policy AIL, we not only require the policy updates to achieve sub-linear regret, but also need the updates to be conservative such that two consecutive policies do not diverge too far away (as stated in Theorem 4.3). Only under this condition, the off-policy updates will not induce excessive distribution shift. In this sense, the regularization in policy update is necessary, and the algorithm we adopted in the main page perfect fits the need, achieving the best known bound in adversarial MDP but also ensures two consecutive policies are close to each other.
>
> > 3. Regarding Eq. (5), it is not clear to me how we can compute this gradient deterministically in practical problems. Perhaps we can get some reliable stochastic estimates of the gradient using samples, but stochastic noise will likely affect the regret bound.
> 4. What does estimation error mean in Theorem 4.4?
>
> We compute the gradient by finite samples, i.e., we use stochastic gradients. The estimation error in Theorem 4.4 is capturing the error of approximating the gradient by finite samples.
>
> > 5. What is the best choice of $N(k)$ that optimizes the upper bound in Theorem 4.4 if it is not restricted to be a constant?
>
> Ideally, the best choice of $N(k)$ depends on the visitation distribution difference between the current policy and previous policies (see Equation 22 in Appendix). At each step, by minimizing the addition of distribution shift error eq 22 and the estimation error eq 26, one can derive the best N(k). However, one does not know the visitation distributions, hence, it is difficult to derive such an instance-dependent bound. Another issue is even though we know the visitation distributions, we can still only derive the optimal N(k) step by step from k=1 to K, and one cannot show it is globally optimal since the choice of N(k) affects the future updates. Nevertheless, we agree deriving an instance-dependent bound would be an interesting future direction.
>
> > 6. Why not include the overall AIL regret bound.
>
> In our original version we do not write the overall regret bound since we think it does not add new insights and is somehow redundant (simply just adding two terms). But we agree including it would benefits the presentation and we have added it in the latest manuscript.

---

> > ### Comment · Reviewer_Uipr · 2026-02-12
> >
> > The authors' response is very clear and has fully addressed my questions. A suggestion regarding bullet 3 and 4 is that the authors should explicitly write Eq. (5) in a way that shows the gradient is estimated using samples.

---

### Review · Reviewer_LxGT · 2026-01-17

**Summary Of Contributions:**

The paper studies convergence and regret guarantees for *off-policy Adversarial Imitation Learning (AIL)* and provides a theoretical analysis showing that reusing data from a limited number of recent policies does not break convergence. While the technical arguments are carefully developed, the overall contribution is significantly limited by the *narrow and outdated problem focus*, as well as by *insufficient experimental validation and positioning with respect to modern imitation learning literature*.

### 1. Overly Narrow and Outdated Scope

The paper focuses exclusively on *off-policy AIL*, a framework that has gradually fallen out of favor in the broader imitation learning (IL) and inverse reinforcement learning (IRL) communities. In recent years, research attention has shifted toward more scalable and stable paradigms, including behavior cloning with regularization, offline RL–based IL, preference-based learning, diffusion-based policy learning, and direct occupancy-matching or value-matching approaches. In contrast, AIL—particularly adversarial formulations inspired by GAIL—has well-known issues with instability, reward identifiability, and sensitivity to discriminator training.

The paper does not convincingly justify *why off-policy AIL remains a problem worth studying today*, beyond historical interest. Without a clear argument for the continued relevance of AIL, the contribution risks appearing incremental and disconnected from the current state of the field.

### 2. Lack of Relevance to State-of-the-Art IL / IRL Methods

A major weakness of the paper is the *absence of comparison—conceptual or empirical—with modern IL and IRL algorithms*. The experimental evaluation compares only against the *on-policy version of the same AIL algorithm* (i.e., varying \(N\) in the proposed method), explicitly stating that comparisons with other off-policy AIL methods or state-of-the-art IL algorithms are omitted.

This raises a fundamental question: *why should the community care about improving off-policy AIL if it is not competitive with, or at least benchmarked against, contemporary alternatives?* Without comparisons to recent IL approaches (e.g., offline RL baselines, non-adversarial IL methods, or modern IRL formulations), the practical significance of the proposed guarantees remains unclear.

### 3. Limited and Unconvincing Experimental Evaluation

The experiments are primarily designed to confirm a narrow theoretical claim: that increasing \(N\) improves sample efficiency relative to the on-policy baseline. While this is consistent with the theory, it does not provide meaningful insight into the algorithm’s competitiveness or usefulness.

- In MiniGrid, the environments are simplistic, the expert demonstrations are hand-crafted, and the comparisons are restricted to variants of the same method.
- In MuJoCo, although neural networks are used, the authors explicitly acknowledge that **their theory does not apply** in this setting, and the implemented loss deviates from the theoretical formulation.

As a result, the experiments do not meaningfully support the broader claim that this approach advances imitation learning practice. Instead, they mainly illustrate a predictable trade-off already suggested by prior empirical work on off-policy data reuse.

### 4. Regret Analysis Largely Inherited from Prior Work

From a theoretical standpoint, much of the regret analysis is built upon existing frameworks, particularly the decomposition introduced by Shani et al. (2022) and related mirror-descent-based analyses. While the paper extends these results to an off-policy setting, the *novelty of the regret bounds appears limited*, with substantial portions of the analysis closely following or adapting prior arguments.

Consequently, the theoretical contribution feels more like a *technical extension within an established framework* rather than a conceptual advance that reshapes how imitation learning problems are approached.

### 5. Overall Assessment

In summary, while the paper is technically sound and well-written, its impact is substantially weakened by:
- an overly narrow focus on off-policy AIL,
- insufficient motivation for revisiting adversarial imitation learning in the current research landscape,
- lack of comparison with modern IL/IRL methods,
- and experimental results that do not convincingly demonstrate practical relevance.

To strengthen the paper, the authors would need to either (i) provide a compelling argument for the renewed importance of AIL, or (ii) reposition the work within a broader imitation learning context, including comparisons to contemporary algorithms and a clearer articulation of what this line of research offers beyond incremental theoretical refinement.

**Audience:**

No

**Audience Explanation:**

Unlikely.

While a small subset of researchers interested in the theoretical analysis of adversarial imitation learning may find the results informative, the topic is narrowly focused on off-policy AIL, which is no longer a central or active area within the broader imitation learning and reinforcement learning communities. As a result, the findings are unlikely to be of interest to a substantial portion of TMLR’s audience.

**Broader Impact Concerns:**

No significant broader impact concerns are identified.

The work is primarily theoretical and methodological, focusing on convergence analysis of a specific imitation learning algorithm. It does not introduce new application domains or deployment scenarios that raise ethical, societal, or safety concerns. The limited practical scope further reduces the likelihood of broader impacts.

**Claims And Evidence:**

No

**Claims Explanation:**

The claims are only partially supported.

The theoretical results are clearly presented and appear correct within the restricted off-policy AIL setting considered. However, the empirical evidence is limited and does not convincingly support broader claims of practical relevance, as experiments only compare against an on-policy variant of the same algorithm and exclude modern imitation learning baselines. In addition, some experiments fall outside the theoretical assumptions, further weakening the overall support for the claims.

**Requested Changes:**

The paper would require substantial revisions to be suitable for publication.

In particular, the authors should
- (i) clearly justify the relevance of adversarial imitation learning in the context of modern imitation and inverse reinforcement learning,
- (ii) include meaningful comparisons with contemporary IL/IRL methods beyond on-policy AIL,
- (iii) strengthen the experimental evaluation to demonstrate practical significance under settings aligned with the theoretical analysis.

Without addressing these points, the contribution remains narrowly scoped and of limited interest.

---

> ### Author Response · Authors · 2026-02-26
>
> We thank the reviewer for their feedback. The main concern from the reviewer is that adversarial imitation learning is no longer a popular paradigm which basically implies any research on AIL is meaningless.
>
> We respectfully disagree with this viewpoint. First, while adversarial imitation learning does face training instability challenges, but simultaneously holds advantages over other paradigms such as automatic representation learning, the ability to learn from limited demonstrations and robustness to long horizons. AIL remains an active and productive area of research. Furthermore, the theoretical understanding and practical techniques developed in one domain frequently guide progresses in other domains — contributions to any rigorous problem have value. Last, more importantly, the principle of TMLR prioritizes technical correctness over subjective assessments of a topic's popularity or significance. Evaluating a work based on whether its research topic is fashionable does not align with these principles.
>
> **We therefore kindly request the reviewer to re-evaluate our work on its own merits: the contributions to broadening the theoretical understanding of AIL, rather than on a comparative assessment of AIL against other paradigms, which falls outside the scope of this work.**
>
> Most of the reviewer's concerns center on the absence of comparisons between AIL and other paradigms. We want to emphasize here that **the focus of this work is to provide a theoretical convergence and sample complexity analysis for a practical variant of AIL algorithms—off-policy AIL which uses past policy data for updates, thereby bridging the gap between theory and practice, instead of introducing a new imitation learning algorithm against other imitation learning paradigms.**
>
> > Overly Narrow and Outdated Scope; Lack of Relevance to State-of-the-Art IL / IRL Methods
>
> We respectfully disagree with the claim that adversarial imitation learning (AIL) is outdated which we have elaborated above. We have updated the related works section to include more discussions of other imitation learning methods. But as we have stated above, the   focus of this work is not introducing a new imitation learning algorithm against other imitation learning paradigms.
>
> > Experiments are limited to variants of the same method and do not demonstrate broader usefulness.
>
> The goal of our experiments is to validate the central theoretical insight: (1). off-policy AIL can converge to a comparable performance to on-policy AIL; (2). showing the effect of the number of reused policies (N) on the performance of off-policy AIL. The experiments serve for our main theorem (Theorem 4.5) and contributions.
>
> > Regret Analysis Largely Inherited from Prior Work
>
> We agree that our work mainly builds on the existing AIL frameworks, but we extend them in a meaningful way: we analyze the off-policy AIL setting which is practically applied and show that data reuse from $o(\sqrt{K})$ recent policies does not break convergence. This is a non-trivial contribution as prior understanding suggested importance sampling might be necessary for off-policy stability. Our result reflects the effect of data reuse in terms of the distribution shift and estimation error, capturing the tradeoff between bias and sample efficiency—an insight not covered in prior work. Our work bridges an existing gap between the theory and practice in AIL.

---

### Review · Reviewer_8ggV · 2026-02-12

**Summary Of Contributions:**

The paper addresses the sample inefficiency of Adversarial Imitation Learning (AIL) by proposing and analyzing a provably efficient Off-Policy AIL algorithm. While standard AIL requires on-policy data for reward updates to maintain convergence guarantees, the authors propose reusing samples from the $N$ most recent policies.
This work develops a unified theoretical and empirical framework for off-policy AIL. It provides a rigorous convergence analysis without relying on importance sampling corrections, decomposing regret into policy regret, reward regret, distribution shift error, and estimation error. The authors prove that when the number of reused past policies grows at a rate of $o(\sqrt{K})$ with respect to the number of iterations, the distribution shift can be controlled and the algorithm converges, offering theoretical support for the common use of replay buffers in practice.

**Audience:**

Yes

**Audience Explanation:**

AIL and GAIL-based methods are widely used in the machine learning community. A persistent issue with these methods is sample inefficiency. Previous work frequently uses replay buffers to mitigate this, often without a solid theoretical basis for convergence stability.

**Claims And Evidence:**

Yes

**Claims Explanation:**

The authors provide comprehensive proofs supporting their main theorems. They utilize standard techniques (Model-based Mirror Descent, Azuma-Hoeffding inequalities) to bound the regret terms. The decomposition of the regret into specific error terms is logically sound and clearly derived. Empirically, the claims are supported by experiments on both tabular and continuous domains. It should be better to contain the baselines in experiments to make a comparison

**Requested Changes:**

In experiments, it should contain some previous works to make a comparison.

---

> ### Author Response · Authors · 2026-02-26
>
> We thank the reviewer for the positive feedback. And we carefully answered your questions below. Hope it solves your concerns.
>
> > In experiments, it should contain some previous works to make a comparison.
>
> The goal of our experiments is to validate the main theoretical results in this work: (1). off-policy AIL can converge to a comparable performance to on-policy AIL; (2). showing the effect of the number of reused policies (N) on the performance of off-policy AIL. The experiments serve for our main theorem (Theorem 4.5) and contributions.
>
> And we want to emphasize that the focus of this work is to provide a theoretical convergence and sample complexity analysis for a practical variant of AIL algorithms—off-policy AIL which uses past policy data for updates, thereby bridging the gap between theory and practice, instead of introducing a new imitation learning algorithm against other imitation learning paradigms, which is outside the scope of our work.

---

### Review · Reviewer_hZrT · 2026-03-25

**Summary Of Contributions:**

This paper proves the convergence properties and sample complexity of off-policy adversarial imitation learning algorithms. It shows that the regret bound of the reward update is sublinear in $K$ when reusing samples from the most recent $o(\sqrt{K})$ policies, under a specific AIL algorithm instance that uses model-based policy mirror descent for policy updates and projected gradient descent for reward updates. The off-policy algorithm achieves better sample efficiency when the estimation error term dominates the error induced by off-policy updates. No importance sampling or other similar off-policy correction techniques is required in the algorithm. The theoretical results are validated through experiments on tabular MiniGrid environments and continuous control MuJoCo environments.

Strengths:

1. The derivation is solid, and the results are useful for understanding the behavior of off-policy adversarial imitation learning algorithms.
2. The paper is clearly written and well-organized.

Weaknesses:

1. The empirical evaluation is limited. While this is understandable given that the paper's primary focus is on providing theoretical foundations for existing off-policy AIL algorithms, the experiments only compare different values of $N$ and the on-policy variant ($N=1$). Including experiments with different base AIL algorithms (e.g., different policy update methods that can also achieve sublinear policy regret) would better demonstrate the applicability of the theoretical findings.
2. There are some inconsistencies in the paper. The $\eta$ value should be $\sqrt{\frac{SA}{K}}$ instead of $\sqrt{\frac{1}{K}}$ in Theorems 4.4 and 4.5. Additionally, before Lemma 4.2, it should be $s \sim \nu^\pi$ rather than $s \sim d^\pi$ since $\nu$ denotes the state visitation distribution. The similar misuse of $d$ and $\nu$ appears multiple times in the paper.

**Audience:**

Yes

**Audience Explanation:**

Readers who work on imitation learning and the broader reinforcement learning domain would be interested in the theoretical results established in this paper.

**Broader Impact Concerns:**

None.

**Claims And Evidence:**

Yes

**Claims Explanation:**

The claims in the paper are supported by clear theoretical derivations. Experiment results on tabular and continuous control environments also empirically validate the effect of different $N$ and state space size on the AIL algorithm.

**Requested Changes:**

1. The inconsistencies mentioned in the weaknesses should be fixed in the paper.
2. More complete experiments with different base AIL algorithms (e.g. other policy update method), should be included to demonstrate that the effect of different $N$ and state space size follows similar trends that align with Theorem 4.5.
3. The current reward update assumes only a single gradient step, which makes the regret bound sensitive to the choice of $\eta$. What will the regret bound be if assuming multiple gradient steps or the exact optimal reward update solution (since $L(\pi, \mu)$ is linear in $\mu$)? Will the regret bound remain the same form only with the $\sqrt{H^2SAK}$ term removed?

---

> ### Author Response · Authors · 2026-04-05
>
> We thank the reviewer for the positive feedbacks and valuable comments on our work. We carefully answered your questions and made corresponding edits. Hope it solved your concerns.
>
> > Including experiments with different base AIL algorithms (e.g., different policy update methods that can also achieve sublinear policy regret) would better demonstrate the applicability of the theoretical findings.
>
> We thank the reviewer for acknowledging that the paper's primary focus is on providing theoretical foundations for existing off-policy AIL algorithms. The current experimental results using model-based policy updates which we introduced in Section 4.1. We conducted additional experiments using policy-gradient updates and provide the result in Appendix D.
>
> > There are some inconsistencies in the paper. The $\eta$ value should be $\sqrt{\frac{SA}{K}}$ instead of $\sqrt{\frac{1}{K}}$ in Theorems 4.4 and 4.5. Additionally, before Lemma 4.2, it should be $s \sim \nu^\pi$ rather than $s \sim d^\pi$ since $\nu$ denotes the state visitation distribution. The similar misuse of $d$ and $\nu$ appears multiple times in the paper.
>
> Thank the reviewer for pointing these out. Yes, the $\eta$ value should be $\sqrt{\frac{SA}{K}}$ and we have corrected it in the updated version. For the inequality before Lemma 4.2, it is state visitation distribution which is the Lemma 3 from Achiam et al. (2017). The corresponding version for state-action visitation distributions in the discounted setting can be derived as: $$\|d^{\pi} - d^{\pi'}\|_1 \le \frac{2}{1-\gamma} \mathbb{E}_{s \sim d^{\pi}} \left[ \mathbb{D}_{\textnormal{TV}}(\pi \| \pi')[s] \right]$$. In Lemma 4.2, we derive the result of state-action visitation distributions in finite-horizon undiscounted MDPs for the purpose of our use.
>
> > The current reward update assumes only a single gradient step, which makes the regret bound sensitive to the choice of $\eta$. What will the regret bound be if assuming multiple gradient steps or the exact optimal reward update solution (since $L(\pi, \mu)$ is linear in $\mu$)? Will the regret bound remain the same form only with the $\sqrt{H^2SAK}$ term removed?
>
> This is an interesting question. First, if we assume the exact optimal reward update solution, then there will be no reward update regret. And since we have the exact optimal solution, there is no need to collect trajectories to estimate the stochastic gradient, and the problem degenerates. The remaining regret reduces to the policy update regret alone. Second, what if we take multiple gradient steps. Note that we cannot simply analyze the optimization error of the reward update at each step, for example, even though the error at each step is $\epsilon$, the summation over k still results in the linear regret in K. The overall analysis of the full regret requires a stable update of $\mu$. And from a practical perspective, for adversarial algorithms, like GAN, over optimization usually deteriorate the stability of training.

---

### Decision · Action_Editor_ABfm · 2026-05-03

**Recommendation:** Accept as is

**Audience:**

Yes

**Audience Explanation:**

The reviewers unanimously agree that the findings are of interest to the TMLR community. It is a suitable contribution for researchers focused on the theoretical foundations of reinforcement learning and imitation learning.

**Claims And Evidence:**

Yes

**Claims Explanation:**

All three reviewers reached a consensus that the claims are supported by rigorous evidence.

Reviewers highlight the paper's contribution in establishing a previously unknown result in AIL and affirms its technical correctness, with "rigorous and welcome theoretical analysis" of off-policy Adversarial Imitation Learning (AIL).

While there was a note regarding the gap between theoretical assumptions and practical application, the core derivations and results were found to be sound.